# Generalizable compound protein interaction prediction with a model incorporating protein structure aware and compound property aware language model representations

Yiming Zhang [1], Ryuichiro Ishitani[2,3,4], Mizuki Takemoto [4] & Atsuhiro Tomita [4] ✉

Compound–protein interaction (CPI) prediction plays a crucial role in drug discovery by aiding the identification of binding and affinities between small molecules and proteins. Current deep learning models rely heavily on sequence-based representations and suffer from a lack of labeled data, which restricts their accuracy and generalizability. To overcome these challenges, we propose GenSPARC (a model with **Gen**eralized **S**tructure- and **P**roperty-**A**ware **R**epresentations of protein and chemical language models for **C**PI prediction), a deep learning model that leverages structure-aware protein representations derived from AlphaFold2 predictions and FoldSeek's three-dimensional interaction alphabet. Compound features were extracted using graph convolutional networks and a pretrained chemical language model, thereby ensuring comprehensive multimodal representation. An attention mechanism further enhanced interaction modeling by capturing intricate binding patterns. GenSPARC was validated successfully with multiple CPI benchmark datasets, demonstrating strong generalizability across challenging data splits and competitive results in virtual screening tasks. Therefore, GenSPARC will substantially advance artificial intelligence-driven drug discovery.

The identification of therapeutically interesting compounds from the estimated $10^{60}$ potential drug-like chemicals remains a critical challenge in drug discovery[1]. High-throughput screening is widely used to evaluate compound–protein binding affinity, but it is time-intensive and costly [2]. Computational methods have emerged as fast and efficient alternatives for predicting compound–protein interaction (CPI)[3–5].

CPI prediction aims to assess how small-molecule compounds interact with proteins by determining their interaction pattern (contact map) and strength (binding affinity)[6]. The accuracy of physics-based CPI prediction methods, such as docking and molecular dynamics simulations, depends heavily on the quality of three-dimensional (3D) protein structures[7,8]. Their performances often decline significantly when predicted rather than experimentally determined structures are used[9,10].

Recent deep learning (DL) models have enabled high-throughput CPI prediction without the need for experimentally determined protein structures, thereby improving the efficiency of binding-affinity estimation[3,11–14]. Early models used 1D convolutional neural networks (CNNs) to represent both compounds and proteins[14,15]; whereas later approaches, such as DeepAffinity[16], adopt recurrent neural networks to process compound and protein sequences. Building on prior developments, GraphDTA[13] incorporates molecular graphs to represent compounds, but protein representations are still limited to sequences. To further refine feature extraction, TransformerCPI[17] and HyperAttentionDTI[18] incorporate self-attention mechanisms. More recently, PerceiverCPI[19] has utilized cross-attention to enhance CPI representations; whereas Cross–Interaction[20] integrates 1D protein sequences with two-dimensional (2D) predicted contact maps for improved binding affinity and contact predictions.

[1]Department of Information and Communications Engineering, School of Engineering, Institute of Science, Tokyo, Yokohama, Kanagawa, Japan. [2]Department of Computational Drug Discovery and Design, Medical Research Laboratory, Institute of Integrated Research, Institute of Science Tokyo, Bunkyo-ku, Tokyo, Japan. [3]Department of Biological Sciences, Graduate School of Science, The University of Tokyo, Bunkyo-ku, Tokyo, Japan. [4]Preferred Networks, Inc., Chiyoda-ku, Tokyo, Japan. ✉e-mail: atomita@preferred.jp

In addition to CPI prediction models, structure-based Drug Target Affinity (DTA) prediction models directly use protein-ligand complex structures or binding pocket information to predict binding affinities[21,22]. These structure-based DTA models require accurate complex structures as input, limiting their applicability when experimentally determined structures are unavailable. Recently, Fold-Docking-Affinity[23] extended a structure-based DTA model by combining AlphaFold2[24] and DiffDock[25] to predict affinities without relying on experimental structures. However, its application results have been limited to kinase targets[23], and the combination of AlphaFold2 structures with DiffDock has shown limited success across diverse targets[26], indicating that extending DTA models to CPI tasks remains difficult.

Despite extensive research on DL-based CPI prediction models, the performance of DL models has been hindered from insufficient generalization to external datasets. When applied beyond the training domain, DL models often fail to maintain predictive accuracy, highlighting a critical limitation of their utility in real-world drug discovery. In the present study, we aimed to address this limitation by addressing two main challenges. First, most models are trained on benchmark CPI datasets, such as Davis[27], KIBA[28], and Metz[29]. These datasets are biased toward a few, well-studied drug target proteins, particularly kinases. Consequently, generalizing DL models trained on these datasets to a broader range of targets is difficult. To address this limitation, You et al.[20] pretrained models on unlabeled databases such as Pfam-A[30] using masked language modeling and graph completion. In addition, they constructed an expanded CPI dataset encompassing a more diverse protein set. Fine-tuning pretrained models on this extended dataset offers a promising approach to overcome protein diversity limitations and extrapolate CPI models to a wider array of drug targets.

The second limitation is the inadequate integration of multimodal information, particularly protein sequences, 3D structural information about proteins, and compound properties, which are all critical for understanding protein functions and their interactions with compounds[31]. With advances in structure prediction models such as AlphaFold2[24], structural information can be utilized for CPI prediction without requiring experimentally determined structures. Previous CPI models have typically treated protein structures, protein sequences, and compound properties as independent sources of information, thereby overlooking the interplay among these modalities[32]. By contrast, the joint representations of PSC–CPI[6], which adopts contrastive machine learning to indirectly capture protein sequences and structures, offer higher generalizability than models relying on sequences or structures alone. Despite its strengths, the representational capacity of PSC–CPI remains insufficient. To enhance both the generalizability and performance of the model, an approach that explicitly incorporates multimodal information, including 3D structural information and compound properties, while leveraging large-scale unlabeled datasets of proteins and compounds, is required. Such an approach enables the learning of more robust embeddings, ultimately improving the accuracy of DL models for CPI prediction and drug discovery[33].

Here, we present a model with **Gen**eralized **S**tructure- and **P**roperty-**A**ware **R**epresentations of protein and chemical language models for **C**PI prediction (GenSPARC). It is designed to integrate multimodal pretrained models capable of incorporating protein sequences, 3D protein structures, and the biochemical features and structures of compounds. GenSPARC adopts the SaProt[34] structure-aware protein language model as a protein encoder, and the Structure-Property Multi-Modal (SPMM)[35] and graph convolutional network (GCN)-integrated model, which captures both structural and biochemical properties, as a compound encoder. To effectively learn joint representations, a multimodal attention network (MAN) was applied. GenSPARC achieved state-of-the-art performance across multiple CPI benchmarks, excelling in binding-affinity and contact-map prediction. Moreover, we performed a benchmark of virtual screening tasks, where prior knowledge of protein-compound complex structures or compound binding pockets is assumed to be known. In the benchmark, our model achieved comparable performance against structure-aware approaches, including DrugCLIP, a state-of-the-art model in virtual screening that explicitly utilizes structural information of protein-compound complex structures. Notably, under the condition that experimental structures were not available and AlphaFold2-predicted structures were used instead, our model demonstrated the best performance.

Compared to other models, GenSPARC demonstrated the best performance in scenarios where actual protein structures were unavailable, highlighting its robustness in experimentally determined, structure-independent CPI predictions.

## Results

### Overview of GenSPARC

As illustrated in Fig. 1, GenSPARC is designed to incorporate both global and local structural information from protein sequences. Using AlphaFold2[24] and Foldseek[36], we predicted the protein structures and assigned a structural alphabet to each residue based on tertiary interactions (Fig. 1A). The obtained structural letters and amino acid sequences were applied to SaProt[34], thereby generating universal, structure-aware protein representations. For the compounds, we converted SMILES representations into molecular graphs, which were processed using GCNs[37] to capture global atomic relationships. These representations were further enriched by 53 structural and biochemical properties extracted using a pretrained property encoder from the SPMM[35] chemical language model (Fig. 1B). To effectively integrate these multimodal representations, we introduced a MAN that captured both intra-modal relationships and inter-modal interactions between proteins and compounds, leveraging large-scale pretrained models based on the Transformer[38] architecture (Fig. 1C).

### Binding affinity and contact prediction on the Karimi dataset

The general CPI prediction framework focuses on two key tasks: contact-map prediction and binding-affinity prediction. In the Karimi dataset setting, prediction is performed using SMILES strings for compounds together with amino acid sequences and AlphaFold2-predicted structures for proteins, without relying on protein-compound complex structural information. In GenSPARC, these tasks were addressed separately by training the dedicated models. In addition, we evaluated the multitask learning version GenSPARC–MT, which employed a combined loss function to learn both tasks simultaneously. Because GenSPARC–MT exhibited similar performance to GenSPARC, the following sections focus primarily on GenSPARC.

We first evaluated our model using four baseline methods, MONN[12], Cross–Interaction[20], PSC–CPI[6], and GraphBAN[39], on the Karimi[11] dataset. MONN[12] is a sequence-based method that jointly predicts binding affinity and atom–residue contacts, representing one of the most effective sequence-based approaches. Cross–Interaction[20] combines protein sequences and structures using 2D contact maps, whereas PSC–CPI[6] integrates sequences and structures through contrastive learning. GraphBAN[39] is a recently reported graph-based framework that integrates inductive learning and knowledge distillation to robustly predict interactions between unseen compounds and proteins. Notably, PSC–CPI[6] is a state-of-the-art model for CPI contact and affinity predictions on the Karimi dataset. The results for the test set were reported across four different data splits following the Original Karimi[11] dataset: (1) *Seen–Both*, with both proteins and compounds present in the training set; (2) *Unseen–Comp*, with proteins present and compounds absent from the training set; (3) *Unseen–Prot*, with proteins not present and compounds absent from the training set; and (4) *Unseen–Both*, with both proteins and compounds absent from in the training set. To evaluate contact-map prediction, we adopted the area under the precision-recall curve (AUPRC) and the area under the receiver operating characteristic curve (AUROC) as metrics. In GraphBAN[39], we followed the analytical procedure described in the original study[39] for contact prediction, where the attention weights of a bilinear attention network were used to predict the contacts between proteins and compounds. For binding-affinity prediction, the root mean square error (RMSE) and Pearson

**A    Protein Sequence Data Construction**

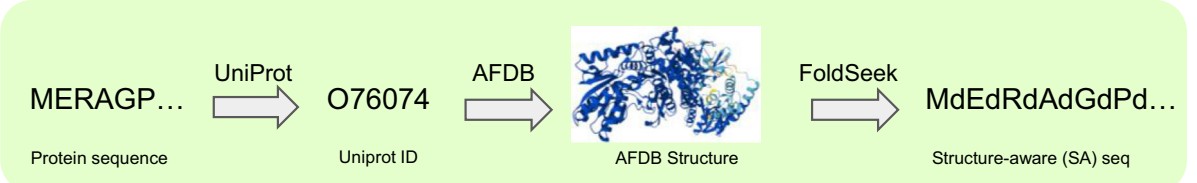

**B    Model Architecture**

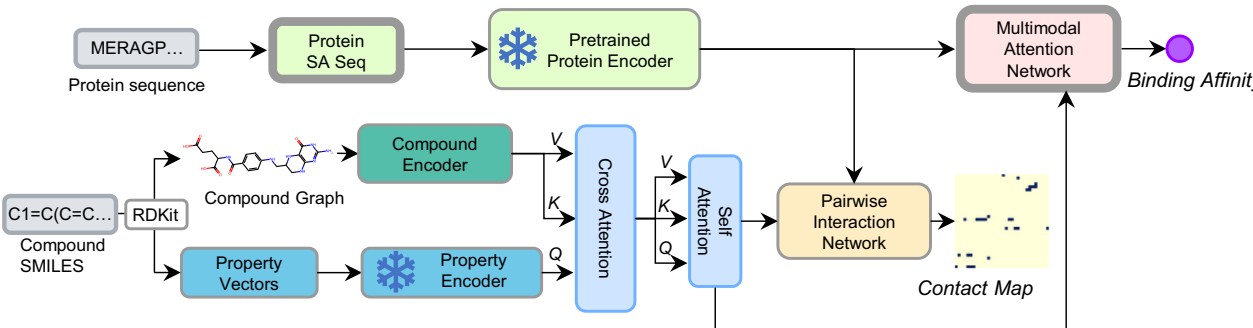

**C    Multimodal Attention Network (MAN)**

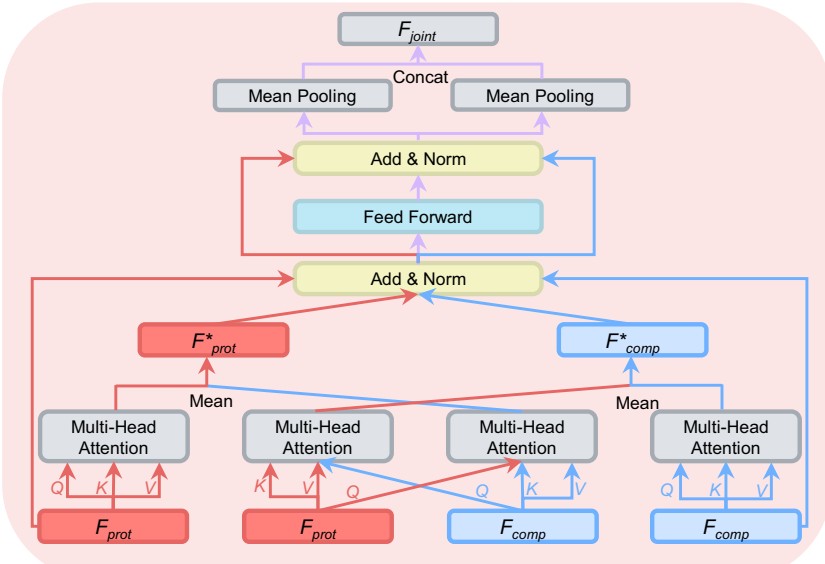

**Fig. 1 | Overall workflow of GenSPARC. A** Generation of protein structure-aware (SA) sequences. Raw protein sequences were first mapped to UniProt IDs and then used to obtain the AlphaFold Protein Structure Database (AFDB) structures. These structures were transformed into SA sequences using FoldSeek. **B** Model architecture. The protein SA sequence was encoded using a pretrained protein encoder, SaProt. Meanwhile, the compound graph, based on SMILES data, was processed through both a compound encoder and a property encoder, capturing structural and chemical features. Cross-attention mechanisms combined the protein and compound data, and self-attention layers further refined these features. The MAN predicted binding affinity, while the pairwise interaction network predicted the contact map. **C** MAN for integrating protein and compound representations. This module combined multi-head self-attention and cross-attention mechanisms, followed by a feed-forward network to capture joint representation between protein residues and compound atoms for predicting binding affinity.

correlation coefficient (PCC) were applied to evaluate the performance of each model. Following the experimental setup of PerceiverCPI[19], we modified the final layers of GraphBAN[39], originally designed for binary classification, to perform regression tasks. These values should be interpreted with caution, as the approach differs from direct prediction methods.

For contact-map prediction, our model outperformed the baseline methods across all splits (Table 1). Specifically, in the Seen–Both test set, our model achieved an AUPRC of 25.44 and an AUROC of 90.54, significantly surpassing the baseline. In the more challenging Unseen–Prot and

Unseen–Both test sets, our model remained robust, with AUPRC of 20.00 and 18.19, respectively. The average performance of our model surpassed that of PSC–CPI[6] by 7.01 in AUPRC and 8.66 in AUROC.

For binding-affinity prediction, our model achieved a lower RMSE and higher PCC than with baseline methods across all splits. Seen–Both achieved an RMSE of 1.357 and a PCC of 0.748. In the Unseen–Comp and Unseen–Prot splits, the model generalized well, with RMSE values of 1.221 and 1.551, and PCCs of 0.787 and 0.526, respectively (Table 2). On average, our model outperformed PSC–CPI[6], showing a 0.096 lower RMSE and

**Table 1 | Performance comparison of CPI on pattern prediction under four data splits on the original Karimi dataset**

| Method | Seen–Both | | Unseen–Comp | | Unseen–Prot | | Unseen–Both | | Average | |
|---|---|---|---|---|---|---|---|---|---|---|
| | AUPRC | AUROC | AUPRC | AUROC | AUPRC | AUROC | AUPRC | AUROC | AUPRC | AUROC |
| MONN | 19.85 | **91.14** | 20.48 | **92.74** | 6.12 | 88.85 | 6.66 | 88.13 | 13.28 | 90.22 |
| Cross–Interaction | 14.46 | 83.20 | 14.19 | 84.27 | 9.54 | 85.34 | 8.99 | 85.79 | 11.80 | 84.65 |
| PSC–CPI | 19.31 | 83.55 | 19.37 | 84.62 | 11.25 | 83.17 | 10.53 | 83.20 | 15.12 | 83.63 |
| GraphBAN | 0.66 | 49.47 | 0.61 | 50.26 | 0.54 | 49.94 | 0.53 | 50.68 | 0.59 | 50.09 |
| GenSPARC | **25.44** | 90.54 | **24.87** | <u>91.57</u> | **20.00** | <u>93.82</u> | **18.19** | <u>93.24</u> | **22.13** | <u>92.29</u> |
| GenSPARC–MT | <u>24.88</u> | <u>90.83</u> | <u>23.98</u> | 91.46 | <u>18.05</u> | **93.91** | <u>17.08</u> | **93.55** | <u>21.00</u> | **92.44** |

Performance comparison of CPI on pattern prediction, measured by AUPRC and AUROC (higher is better), under four data splits on the original Karimi dataset. The best and second-best metrics are marked with bold and underline, respectively.

**Table 2 | Performance comparison of CPI on binding-affinity prediction under four data splits on the original Karimi dataset**

| Method | Seen–Both | | Unseen–Comp | | Unseen–Prot | | Unseen–Both | | Average | |
|---|---|---|---|---|---|---|---|---|---|---|
| | RMSE↓ | PCCs↑ | RMSE↓ | PCCs↑ | RMSE↓ | PCCs↑ | RMSE↓ | PCCs↑ | RMSE↓ | PCCs↑ |
| MONN | 1.418 | 0.710 | 1.314 | 0.737 | 1.700 | 0.420 | 1.690 | 0.501 | 1.531 | 0.553 |
| Cross–Interaction | 1.570 | 0.631 | 1.397 | 0.684 | 1.662 | 0.463 | 1.762 | 0.430 | 1.598 | 0.552 |
| PSC–CPI | 1.472 | 0.690 | 1.350 | 0.712 | 1.613 | <u>0.480</u> | <u>1.630</u> | <u>0.560</u> | 1.516 | <u>0.611</u> |
| GraphBAN | 1.939 | 0.340 | 1.775 | 0.459 | 1.661 | 0.378 | 1.770 | 0.373 | 1.786 | 0.387 |
| GenSPARC | **1.357** | <u>0.748</u> | **1.221** | **0.787** | **1.551** | **0.526** | **1.552** | **0.583** | **1.420** | **0.661** |
| GenSPARC–MT | <u>1.363</u> | **0.753** | <u>1.270</u> | <u>0.760</u> | 1.620 | 0.470 | 1.802 | 0.395 | <u>1.514</u> | 0.595 |

Performance comparison of CPI on binding-affinity prediction with rooted mean square error (RMSE, lower is better) and Pearson correlation coefficients (PCCs, higher is better) under four data splits on the original Karimi dataset. The best and second-best metrics are marked with bold and underline, respectively.

0.050 higher PCC, thereby demonstrating strong performance and generalizability.

## Binding affinity and contact prediction on a split test set to evaluate sequence and structural similarities

While we expected worse performance on the Unseen–Comp split set compared to the Seen–Both split set, because the former included compounds absent from the training data, the results showed comparable or even slightly better performance (Table 2). This unexpected outcome raised concerns about potential data leakage across different splits. To address this issue, we implemented a more stringent dataset splitting approach. Strict Tanimoto similarity thresholds were enforced for the compounds, while two distinct split settings were introduced for proteins: a sequence-hard setting based on sequence alignment scores and a structure-hard setting based on structural alignment scores.

Using both settings, it was possible to more comprehensively evaluate the model's generalizability. The results were consistent with our expectations (Tables 3 and 4). In the sequence-hard setting, all direct prediction models (MONN[12], Cross–Interaction[20], PSC–CPI[6], and GenSPARC) showed improved contact prediction performance (Table 3), with the Seen–Both test set achieving the best results. The MONN[12], Cross–Interaction[20], PSC–CPI[6], and GraphBAN[39] models exhibited an expected decrease in performance for the Unseen–Prot test set and Unseen–Both test set. In contrast, GenSPARC demonstrated superior performance than the baseline models, with notable improvements in the Unseen–Prot and Unseen–Both test sets that pointed to robust generalizability, even under stringent split conditions.

Similar results were obtained with the structure-hard setting (Table 3). While the baseline models showed reduced performance in the structure-hard Unseen–Prot and Unseen–Both configurations, GenSPARC improved its performance in the Unseen–Prot split and maintained a consistent effectiveness in the Unseen–Both split. Our model substantially

outperformed PSC–CPI[6], achieving remarkable gains of +7.82 in AUPRC and +9.13 in AUROC for sequence-hard evaluations, alongside improvements of +7.08 in AUPRC and +8.86 in AUROC for structure-hard assessments. Although GenSPARC showed a slightly worse performance in Unseen–Comp across both sequence-hard and structure-hard configurations, these changes were minimal. Moreover, for binding-affinity prediction (Table 4), GenSPARC generally outperformed all baseline models across all evaluation settings, further validating its robustness under diverse and challenging conditions.

In conclusion, these comprehensive results demonstrate the superiority of GenSPARC for contact-map prediction and binding-affinity estimation. Our model outperformed standard evaluation metrics and, more importantly, maintained exceptional performance under rigorous evaluation frameworks specifically designed to test true generalizability.

## Case studies for CPI patterns

To evaluate the performance and generalization of GenSPARC, we examined its ability to reconstruct contact maps for two protein examples: prothrombin (UniProt ID: P00734) and macrophage metalloelastase (UniProt ID: P39900). Prothrombin represented the Unseen–Prot split, whereas macrophage metalloelastase represented the more challenging Unseen–Both split. We compared the predicted contact maps of GenSPARC with those of Cross–Interaction[20] and PSC–CPI[6] to assess the ability of the models to recover ground-truth interaction patterns. To ensure a fair evaluation, we ranked the interaction strengths, set a threshold to match the number of ground-truth interactions, and normalized the results (Fig. 2).

GenSPARC demonstrated superior predictive accuracy for prothrombin than the baseline methods. Cross–Interaction[20] failed to identify key interaction sites in the bottom-left region (corresponding to Val213–Cys220 in the protein and atoms 1–6 in the compound) or in the bottom-right corner (corresponding to Gly226 in the protein and atom 16 in the compound). While PSC–CPI[6] showed partial recovery of the bottom-

**Table 3 | Performance comparison of CPI on pattern prediction under four data splits on the sequence-hard and structure-hard dataset settings**

| Method | Seen–Both | | Unseen–Comp | | Unseen–Prot | | Unseen–Both | | Average | |
|---|---|---|---|---|---|---|---|---|---|---|
| | AUPRC | AUROC | AUPRC | AUROC | AUPRC | AUROC | AUPRC | AUROC | AUPRC | AUROC |
| Contact prediction under the sequence-hard setting | | | | | | | | | | |
| MONN | 27.21 | **94.94** | **15.36** | **92.92** | 4.46 | 85.66 | 5.13 | 85.74 | 13.04 | 89.82 |
| Cross–Interaction | 18.10 | 85.41 | 10.00 | 86.55 | 8.57 | 80.99 | 8.10 | 84.59 | 11.19 | 84.39 |
| PSC–CPI | 28.99 | 87.25 | 10.91 | 86.09 | 10.20 | 79.76 | 8.34 | 80.14 | 14.61 | 83.31 |
| GraphBAN | 0.59 | 49.54 | 0.39 | 50.63 | 0.95 | 52.00 | 0.56 | 52.20 | 0.623 | 51.09 |
| GenSPARC | **33.25** | 91.52 | <u>12.60</u> | <u>91.21</u> | **25.34** | <u>92.89</u> | **18.54** | <u>94.12</u> | **22.43** | <u>92.44</u> |
| GenSPARC–MT | <u>32.06</u> | 91.98 | 11.71 | 91.05 | <u>25.01</u> | **93.64** | 15.10 | **94.14** | <u>20.97</u> | **92.70** |
| Contact prediction under the structure-hard setting | | | | | | | | | | |
| MONN | 24.55 | **94.77** | **13.06** | **91.69** | 5.82 | 85.21 | 4.96 | 87.31 | 12.10 | 89.75 |
| Cross–Interaction | 17.43 | 85.59 | 8.85 | 85.21 | 8.68 | 80.14 | 8.10 | 84.94 | 10.77 | 83.97 |
| PSC–CPI | 27.98 | 87.63 | 9.82 | 84.87 | 10.60 | 79.44 | 8.07 | 80.79 | 14.12 | 83.18 |
| GraphBAN | 0.57 | 48.00 | 0.40 | 48.88 | 0.79 | 51.79 | 0.56 | 51.8 | 0.58 | 50.12 |
| GenSPARC | <u>32.09</u> | <u>91.86</u> | <u>11.97</u> | <u>91.09</u> | 23.35 | **91.27** | **17.38** | <u>93.92</u> | <u>21.20</u> | **92.04** |
| GenSPARC–MT | **32.72** | 91.69 | 11.57 | 90.99 | **23.75** | <u>91.24</u> | <u>16.78</u> | **93.44** | **21.21** | <u>91.84</u> |

Performance comparison of CPI on pattern prediction, measured by AUPRC and AUROC (higher is better), under four data splits on the sequence-hard and structure-hard dataset settings. The best and second-best metrics are marked with bold and underline, respectively.

**Table 4 | Performance comparison of CPI on binding-affinity prediction under four data splits on the sequence-hard and structure-hard dataset settings**

| Method | Seen–Both | | Unseen–Comp | | Unseen–Prot | | Unseen–Both | | Average | |
|---|---|---|---|---|---|---|---|---|---|---|
| | RMSE↓ | PCCs↑ | RMSE↓ | PCCs↑ | RMSE↓ | PCCs↑ | RMSE↓ | PCCs↑ | RMSE↓ | PCCs↑ |
| Affinity prediction under the *Sequence Hard* setting | | | | | | | | | | |
| MONN | 1.219 | 0.761 | **1.314** | **0.648** | <u>1.655</u> | <u>0.491</u> | **1.553** | 0.500 | <u>1.435</u> | <u>0.600</u> |
| Cross–Interaction | 1.451 | 0.623 | 1.575 | 0.455 | 1.776 | 0.463 | 1.806 | 0.470 | 1.652 | 0.503 |
| PSC–CPI | 1.280 | 0.736 | 1.507 | 0.510 | 1.695 | 0.490 | 1.714 | **0.528** | 1.549 | 0.566 |
| GraphBAN | 1.433 | 0.707 | 1.388 | 0.614 | 1.874 | 0.440 | 1.615 | 0.500 | 1.578 | 0.565 |
| GenSPARC | <u>1.121</u> | <u>0.805</u> | <u>1.372</u> | 0.631 | **1.444** | **0.642** | <u>1.624</u> | <u>0.522</u> | **1.390** | **0.650** |
| GenSPARC–MT | **1.063** | **0.825** | 1.380 | <u>0.643</u> | 1.722 | 0.476 | 1.651 | 0.327 | 1.454 | 0.568 |
| Affinity prediction under the *Structure Hard* setting | | | | | | | | | | |
| MONN | 1.250 | 0.733 | 1.442 | 0.605 | <u>1.600</u> | <u>0.551</u> | **1.427** | **0.570** | 1.430 | 0.615 |
| Cross–Interaction | 1.425 | 0.660 | 1.673 | 0.403 | 1.696 | 0.479 | 1.736 | 0.498 | 1.633 | 0.510 |
| PSC–CPI | 1.406 | 0.652 | 1.569 | 0.483 | 1.712 | 0.456 | 1.547 | 0.477 | 1.559 | 0.517 |
| GraphBAN | 1.424 | 0.696 | 1.441 | 0.612 | 1.761 | 0.479 | 1.620 | 0.522 | 1.562 | 0.578 |
| GenSPARC | <u>1.141</u> | <u>0.793</u> | <u>1.411</u> | <u>0.618</u> | **1.519** | **0.582** | 1.559 | <u>0.546</u> | <u>1.407</u> | **0.635** |
| GenSPARC–MT | **1.104** | **0.802** | **1.297** | **0.675** | 1.636 | 0.501 | <u>1.521</u> | 0.521 | **1.390** | <u>0.625</u> |

Performance comparison of CPI on binding-affinity prediction with rooted mean square error (RMSE, lower is better) and Pearson correlation coefficients (PCCs, higher is better) under four data splits on the sequence-hard and structure-hard dataset settings. The best and second-best metrics are marked with bold and underline, respectively.

left interaction, it failed to identify contacts in the bottom-right region. In contrast, GenSPARC successfully identified both interaction regions and recovered the interaction at G226. For macrophage metalloelastase, all three models identified the upper-right interaction site (corresponding to Gly179–Ala182 in the protein and atoms 17–20 in the compound). However, noticeable differences were observed for the other contact regions. Both Cross–Interaction[20] and PSC–CPI[6] failed to recover the upper-left interaction (Gly179–Ala182 in the protein and atoms 1–8 in the compound) and the bottom interaction site (Tyr240 in the protein and atoms 13–16 in the compound). Although GenSPARC exhibited a slightly weaker signal in these regions, it successfully detected both contact sites, demonstrating an improved ability to identify CPI in structurally difficult cases.

## Additional binding-affinity prediction using Davis, KIBA, and Metz datasets

Next, we compared GenSPARC and previous models in terms of CPI binding-affinity prediction using three public datasets—Davis[27], KIBA[28], and Metz[29]—under the sequence-hard Seen–Both, Unseen–Comp, Unseen–Prot, and Unseen–Both split settings. The mean squared error (MSE) and concordance index were used as evaluation metrics. We compared GenSPARC to DeepConvDTI[15], GraphDTA[13], HyperattentionDTI[18], TransformerCPI[17], PerceiverCPI[19], Cross–Interaction[20], PSC-CPI[6], and GraphBAN[39]. We followed the experimental setup of PerceiverCPI[19], modifying the final layers of binary classification models (e.g., TransformerCPI, DeepConvDTI, Hyper-AttentionDTI, and GraphBAN) for the regression tasks.

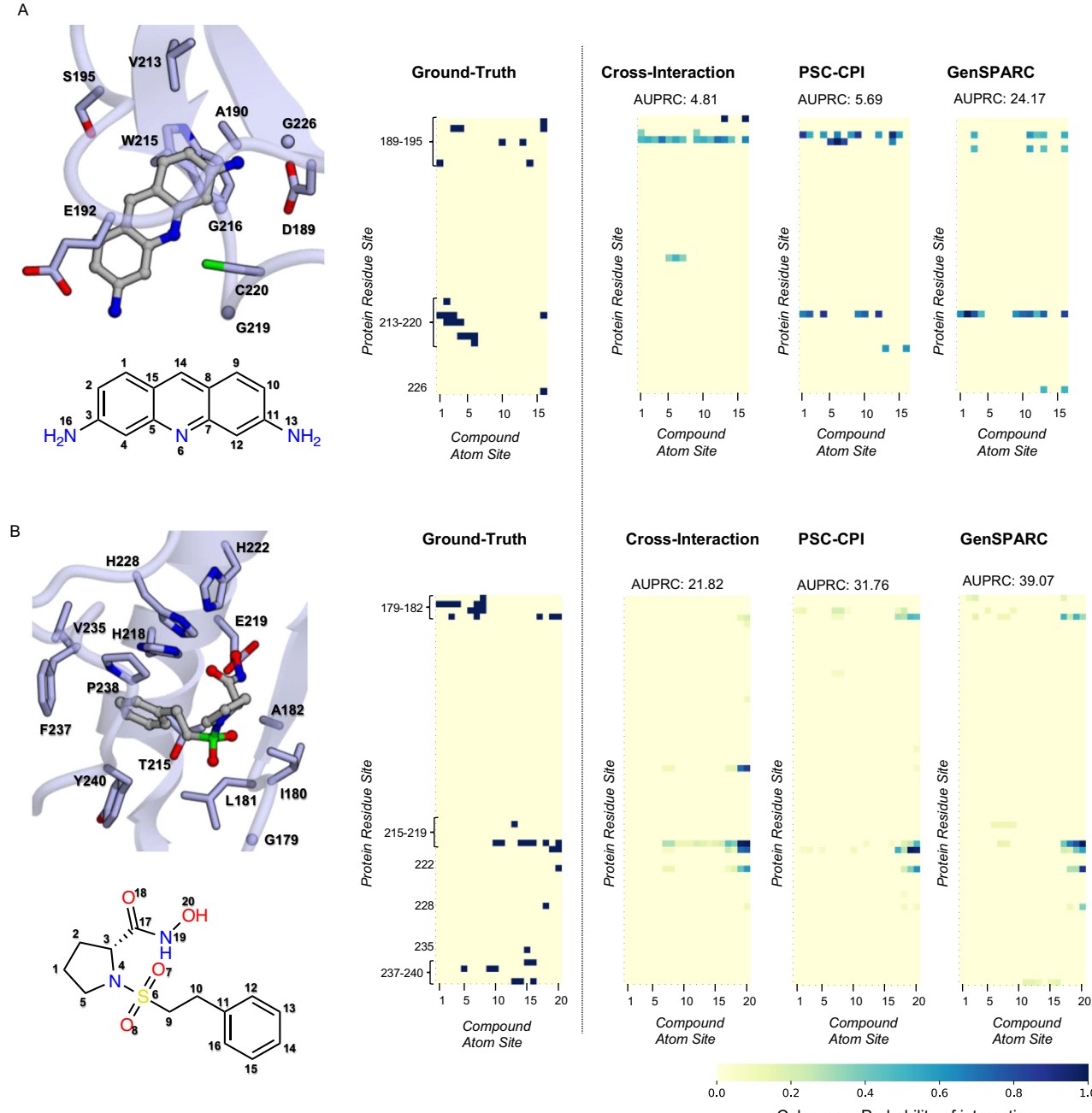

**Fig. 2 | Comparison of predicted CPI contact maps across different models.**
**A**, **B** For two representative protein-compound complexes, the interaction sites of their experimentally determined structures and structural formula of the compound, the ground-truth contact maps, and the predicted contact maps from Cross–Interaction, PSC–CPI, and GenSPARC are illustrated from left to right. In the structure shown in the left panel, the compound is depicted using a ball-and-stick representation, while the side chains and Cα atoms of the interacting amino acids are represented as sticks. Protein structures and compound structural formulae are annotated with residue numbers and atomic numbers corresponding to the contact map. The contact maps depict interactions between residue sites (proteins) and atomic sites (compounds), with darker hues indicating higher prediction accuracy. The AUPRC scores (higher is better) are displayed for each method. Prothrombin (PDB: 1BCU; **A**) and macrophage metalloelastase (PDB: 3RTT; **B**) were used to generate the displayed experimental structures and ground-truth contact maps.

For clarity, we focused on the Unseen–Both split setting shown in Table 5, as it is the most important setting for assessing generalizability, whereas the results for all splits (Seen–Both, Unseen–Comp, Unseen–Prot, and Unseen–Both) are provided in the Supplementary Information (Tables S1–S3). GenSPARC exhibited consistently superior performance across the most unseen test settings, demonstrating strong generalization capability beyond the training distribution (Table 5, Tables S2 and S3). Under the Unseen–Both split, both Cross–Interaction and PSC–CPI[6] demonstrated competitive MSE performance but were slightly outperformed by PerceiverCPI in terms of concordance index (Table 5). Notably, GenSPARC achieved superior MSE results across all datasets, delivering the best performance in four of six metrics and underscoring its effectiveness in CPI prediction under stringent evaluation conditions.

## Virtual screening using the DUD-E benchmark
We assessed the performance of GenSPARC in virtual screening tasks using the Database of Useful Decoys-Enhanced (DUD-E) benchmark. Thus, we

**Table 5 | Performance comparison of GenSPARC with state-of-the-art baselines for CPI strength prediction**

| Method | Davis | | KIBA | | Metz | |
|---|---|---|---|---|---|---|
| | MSE↓ | CIndex↑ | MSE↓ | CIndex↑ | MSE↓ | CIndex↑ |
| DeepConvDTI | 2.065 ± 0.255 | 0.581 ± 0.080 | 5.944 ± 0.348 | 0.531 ± 0.020 | 2.098 ± 0.182 | 0.523 ± 0.010 |
| GraphDTA(GINs) | 1.200 ± 0.770 | 0.481 ± 0.035 | 0.713 ± 0.598 | 0.512 ± 0.073 | 0.768 ± 0.181 | 0.514 ± 0.024 |
| HyperattentionDTI | 1.450 ± 1.051 | 0.568 ± 0.066 | 2.602 ± 0.504 | 0.540 ± 0.035 | 0.877 ± 0.394 | 0.513 ± 0.085 |
| TransformerCPI | 1.076 ± 0.764 | 0.487 ± 0.041 | <u>0.648 ± 0.124</u> | 0.529 ± 0.015 | 0.777 ± 0.726 | 0.518 ± 0.031 |
| PeceiverCPI | 0.965 ± 0.164 | **0.604 ± 0.018** | 1.925 ± 0.499 | <u>0.557 ± 0.026</u> | 1.448 ± 0.510 | 0.533 ± 0.034 |
| Cross–Interaction | 0.701 ± 0.236 | 0.544 ± 0.036 | 0.699 ± 0.099 | 0.512 ± 0.022 | 0.574 ± 0.138 | 0.515 ± 0.032 |
| PSC–CPI | <u>0.698 ± 0.248</u> | 0.579 ± 0.046 | 0.676 ± 0.041 | 0.532 ± 0.032 | <u>0.526 ± 0.059</u> | 0.510 ± 0.066 |
| GraphBAN | 0.804 ± 0.111 | 0.582 ± 0.021 | 0.713 ± 0.054 | **0.580 ± 0.011** | 0.651 ± 0.053 | <u>0.547 ± 0.032</u> |
| GenSPARC | **0.682 ± 0.224** | <u>0.588 ± 0.039</u> | **0.630 ± 0.034** | 0.544 ± 0.028 | **0.509 ± 0.046** | **0.570 ± 0.033** |

Performance comparison of GenSPARC with other state-of-the-art baselines for CPI strength prediction on three public datasets under the Unseen–Both sequence-hard setting. The best and second-best metrics are marked with bold and underline, respectively. Data are presented as the mean ± SD of five-fold cross-validation.

**Table 6 | Virtual screening on DUD-E in the zero-shot setting**

| | AUROC (%) | BEDEOC (%) | EF | | |
|---|---|---|---|---|---|
| | | | 0.5% | 1% | 5% |
| PDB structures as input | | | | | |
| Glide-SP* | <u>76.70</u> | <u>40.70</u> | <u>19.39</u> | <u>16.18</u> | <u>7.23</u> |
| Vina–PDB* | 71.60 | - | 9.13 | 7.32 | 4.44 |
| NN-score* | 68.30 | 12.20 | 4.16 | 4.02 | 3.12 |
| RF-score* | 65.21 | 12.41 | 4.90 | 4.52 | 2.98 |
| Pafnucy* | 63.11 | 16.50 | 4.24 | 3.86 | 3.76 |
| OnionNet* | 59.71 | 8.62 | 2.84 | 2.84 | 2.20 |
| Planet* | 71.60 | - | 10.23 | 8.83 | 5.40 |
| DrugCLIP–PDB* | **80.93** | **50.52** | **38.07** | **31.89** | **10.66** |
| GenSPARC–PDB | 74.60 | 22.32 | 15.10 | 12.82 | 6.31 |
| AlphaFold2 structures as input | | | | | |
| Vina–AF | **64.30** | - | <u>4.63</u> | <u>4.08</u> | <u>2.81</u> |
| DrugCLIP–AF | 52.63 | <u>3.61</u> | 2.12 | 1.93 | 1.40 |
| GenSPARC–AF | <u>60.46</u> | **11.46** | **7.14** | **6.59** | **3.47** |

Virtual screening results on DUD-E in the zero-shot setting. Results are reported for AUROC, BEDROC, and enrichment factor (EF) values at different thresholds. Values marked with * indicate results obtained from the DrugCLIP article[12]. The best and second-best metrics are marked with bold and underline, respectively.

benchmarked GenSPARC against a diverse set of approaches, including conventional docking methods, machine learning models, and DL approaches (Table 6). In contrast to the CPI prediction task discussed above, this benchmark includes models that utilize experimentally determined complex structures and binding pocket information provided in the dataset. In the zero-shot setting, GenSPARC was benchmarked against the docking methods Glide-SP[40] and AutoDock Vina[7], the machine learning models random score (RF-score)[41], and the DL models NNScore[42], Pafnucy[43], OnionNet[44], Planet[45], and the current state-of-the-art model DrugCLIP[46]. In the cross-validation setting, GenSPARC was compared to RF-score[41], NNScore[42], AutoDock Vina[7], and the DL models 3D-CNN[47], Graph CNN[2], COSP[48], DrugVQA[49], and the state-of-the-art methods AttentionSiteDTI[50] and DrugCLIP[46].

Conventional docking and structure-aware models, which rely heavily on experimentally determined protein-ligand pocket coordinates, often struggle to generalize to predicted structures. To systematically evaluate this limitation, we tested two versions of AutoDock Vina[7], a representative docking model, and DrugCLIP, a representative structure-aware model, under a zero-shot setting, while maintaining the same model architecture but altering the input data source. Specifically, we evaluated *Vina–PDB*, using

coordinates from experimentally determined Protein Database (PDB) structures as input, and *Vina–AF*, using coordinates from AlphaFold2 structures as input. Similarly, we assessed *DrugCLIP–PDB*, using pocket coordinates from experimentally determined PDB structures as input, and *DrugCLIP–AF*, using pocket coordinates from AlphaFold2 structures as input. This comparison offered insights into the ability of AutoDock Vina and DrugCLIP to generalize experimental structures and predicted protein structures, which is a key challenge in structure-based virtual screening. Similarly, we evaluated GenSPARC in two configurations: *GenSPARC–PDB*, using FoldSeek's structural alphabet generated from experimentally determined PDB structures as inputs, and *GenSPARC–AF*, using FoldSeek's structural alphabet generated from AlphaFold2 structures as inputs.

For cross-validation, GenSPARC used structure-aware sequences derived from experimentally determined PDB structures, following the data splitting and evaluation method of AttentionSiteDTI[50]. Although AUROC is a widely used metric for classification, it has been criticized for its limitations in virtual screening, as it does not prioritize the early retrieval of active compounds[51,52]. Therefore, we also employed the enrichment factor (EF)[53], Boltzmann-enhanced discrimination of ROC (BEDROC)[51], and ROC enrichment (RE)[54,55] metrics to better capture early enrichment performance and the actual effectiveness of virtual screening models.

DrugCLIP–PDB achieved strong performance, with a BEDROC ($\alpha = 85$) of 50.52 and EF values of 38.07 (EF 0.5%), 31.89 (EF 1%), and 10.66 (EF 5%) (Table 6). Instead, DrugCLIP–AF dropped to an EF value of 2.12 (EF 0.5%), indicating its dependence on precise experimental structures. In contrast, GenSPARC–PDB achieved competitive results compared with DrugCLIP–PDB, with a BEDROC ($\alpha = 85$) of 22.32 and EF values of 15.10 (EF 0.5%) and 12.82 (EF 1%). Focusing on the results using AlphaFold2 structures as input, all evaluated methods, including GenSPARC, DrugCLIP, and Vina, exhibited a performance drop compared with their performance using experimentally determined PDB structures, highlighting the limitations of predicted structures. Notably, GenSPARC–AF still maintained a robust performance, reaching an EF value of 7.14 (EF 0.5%), achieving the highest EF value among the evaluated methods, including DrugCLIP–AF and Vina–AF. These findings indicate that GenSPARC maintained superior performance in virtual screening scenarios where experimentally determined structures are unavailable.

In the cross-validation setting, GenSPARC demonstrated competitive performance with state-of-the-art methods across all evaluation metrics, with a strong ability to identify hit molecules within a small fraction of the dataset (Table 7).

Overall, GenSPARC exhibited a robust performance in virtual screening tasks, whether using experimentally determined structures or using predicted structures. Notably, GenSPARC showed strong performance in hit identification metrics, highlighting its potential for practical applications in structure-based virtual screening.

**Table 7 | Virtual screening on DUD-E in the cross-validation setting**

| | AUROC (%) | RE | | | |
|---|---|---|---|---|---|
| | | 0.5% | 1% | 2% | 5% |
| Vina* | 69.60 | 9.139 | 7.321 | 5.811 | 4.444 |
| NN-score* | 58.40 | 4.166 | 2.980 | 2.460 | 1.891 |
| RF-score* | 0.622 | 5.628 | 4.274 | 3.499 | 2.678 |
| 3D-CNN* | 86.80 | 42.56 | 26.66 | 19.36 | 10.71 |
| Graph CNN* | 88.60 | 44.41 | 29.75 | 19.41 | 10.74 |
| COSP* | 90.10 | 51.05 | 35.98 | 23.68 | 12.21 |
| DrugVQA* | **97.20** | 87.77 | 58.51 | 34.48 | **17.38** |
| AttentionSiteDTI* | <u>97.10</u> | 101.74 | <u>59.92</u> | <u>35.07</u> | 16.74 |
| DrugCLIP* | 96.59 | **118.10** | **67.17** | **37.17** | 16.59 |
| GenSPARC | 94.93 | <u>106.71</u> | 59.67 | 33.37 | 15.37 |

Virtual screening results on DUD-E in the cross-validation setting. Results are reported for AUROC and ROC enrichment metric (RE) values at different thresholds. Values marked with * indicate results obtained from the AttentionSiteDTI article[42] and the DrugCLIP article[12]. The best and second-best metrics are marked with bold and underline, respectively.

## Structure and property representations enhance generalizability

To evaluate the impact of structure- and property-aware representations on model performance, we conducted an ablation study comparing the full GenSPARC model with three simplified variants: (A) without protein structure awareness, using raw protein sequences processed by ESM-2 650M[56] embeddings; (B) without the GCN, utilizing only SMILES-based transformer embeddings[35] for compound representations; and (C) without property encoder, omitting 53 additional molecular properties (Table S4). These configurations were evaluated under the Unseen–Both split, which is the most challenging scenario for generalization. The results for the sequence-hard setting are shown in Fig. 3, while detailed results across all settings, including the original Karimi dataset, sequence-hard, and structure-hard configurations, are provided in Fig. S3. Results for all the splits, including Seen–Both split, Unseen–Comp, Unseen–Prot, and Unseen–Both split, are presented in Tables S5 and S6.

In binding-affinity prediction, the full GenSPARC model achieved the best results, yielding the lowest RMSE and highest PCC (Fig. 3A, B). Ablation studies revealed that removing any single component, be it structure awareness, GCN, or property encoder, led to noticeable performance declines, highlighting the necessity of each for predictive accuracy and generalization.

For contact prediction, the full GenSPARC model consistently outperformed the alternatives across all splits. Although removing structure awareness or the property encoder had a minor impact on the original Karimi dataset, it led to significantly worse performance in the sequence-hard and structure-hard settings, emphasizing their role in enhancing model discrimination and generalization (Fig. 3 and S3). Notably, removing the GCN caused a sharp decline to an AUPRC of 10–11 (Fig. 3C), whereas AUROC (Fig. 3D) remained comparable to that in the full model. This indicates that the GCN plays a critical role in reducing false positives, which is crucial for accurate contact prediction.

Overall, the results from the Unseen–Both split highlight the critical role of integrating multiple components—structure awareness, graph-based molecular representations, and molecular properties—to achieve strong generalization in both contact and affinity prediction tasks. While structure awareness and the GCN had the largest impact on performance, the molecular property encoder provided valuable complementary information to further enhance the effectiveness of the model.

## Discussion

In this study, we describe GenSPARC, a model designed to enhance CPI prediction. GenSPARC achieves this goal by integrating global and local 3D

structural information from protein sequences. By leveraging AlphaFold2 to predict protein structures, FoldSeek to assign structural letters to residues, and SaProt to generate embeddings, GenSPARC overcomes the limitations of previous methods, which were restricted to local pockets. Additionally, the model incorporates the structural and biochemical properties of compounds through GCNs and a multimodal molecular model, significantly boosting its predictive performance. Extensive experiments demonstrated that GenSPARC achieved state-of-the-art performance across multiple benchmarks, including affinity and contact map prediction tasks, on established datasets (i.e., Karimi, Davis, KIBA, and Metz), even under different splitting criteria. Notably, our stricter dataset splitting demonstrated the strong generalizability of GenSPARC, while revealing the shortcomings of existing models, which struggled to perform consistently under more realistic conditions. These results underscore the need for benchmarks and datasets that better reflect the challenges of drug discovery and provide more accurate assessments of model capabilities. GenSPARC performed comparably well on the DUD-E virtual screening benchmark, demonstrating its robustness in actual applications. These findings highlight the effectiveness of GenSPARC in overcoming the structural limitations of previous methods and tackling the dynamic complexities of CPI. Nonetheless, certain limitations and challenges remain, and they provide directions for further improvement.

First, the structure and property encoder from SPMM uses primarily high-level molecular descriptors from RDKit, such as aromaticity, Gasteiger charges, and ring counts. Although effective for basic molecular properties, these features may not fully capture the complexity of 3D compound structures. Notably, in the sequence and structure-hard settings, AUPRC under the Unseen–Comp split showed significantly worse contact prediction (Table 3), whereas the Unseen–Prot split demonstrated stability and even improvements. This indicates that the compound encoder can be improved further. One potential approach is to incorporate more 3D compound information, such as spatial embeddings, to enhance performance. However, the direct use of absolute distances or coordinates may not be ideal because it can lead to poor generalization. They tend to overfit the experimental data and struggle to perform well on predicted structures such as those from AlphaFold2. A more robust approach would be to develop representations that remain consistent across both experimental and predicted structures. Developing robust representations, such as FoldSeek's structural letters, which are consistent across predicted and experimental structures, could address this challenge. Second, although GenSPARC improved accuracy, some contact prediction results were still missing (Fig. 2). A potential strategy is to use predicted interactions as constraints in complex structure prediction models such as AlphaFold3[57] or Boltz-1[58], enabling the generation of plausible complex structures consistent with GenSPARC's predictions, which can then be further refined to the level of molecular dynamics. Third, a related challenge concerns the balance between accuracy and generalization in structural representations. Although structure-aware models such as DrugCLIP achieved higher peak accuracy by directly using 3D coordinates, our use of FoldSeek's 3Di representation inevitably sacrificed some fine-grained structural information. Nevertheless, FoldSeek's structural alphabet has been shown through mutual information analyses to retain richer features than prior structural encodings, and, importantly, the discretization of local structures provides robustness against coordinate noise and enhances generalizability, as demonstrated in our results. Bridging this trade-off by designing representations that simultaneously preserve high accuracy while ensuring robust generalization remains an important avenue for future research. One potential direction is the development of integrated pipelines, such as the recently introduced Boltz-2[59], which jointly perform structure prediction and affinity prediction, aiming for a more seamless integration of structural modeling with compound–protein interaction prediction. Comprehensive benchmarking, including CPI prediction models integrated with structure prediction pipelines, as well as further model development, will be essential to advance the field. Finally, another limitation of this work is the absence of intrinsically disordered proteins (IDPs) in the training datasets. Future work

**Fig. 3 | Impact of structure awareness, GCN, and property encoder on model performance under sequence-hard settings. A** RMSE, (**B**) PCC, (**C**) AUPRC, and (**D**) AUROC metrics were evaluated for the full GenSPARC model and three alternative configurations: without structure awareness, without GCN, and without the property encoder. Data are presented as the mean ± SD over three independent experiments with different random seeds.

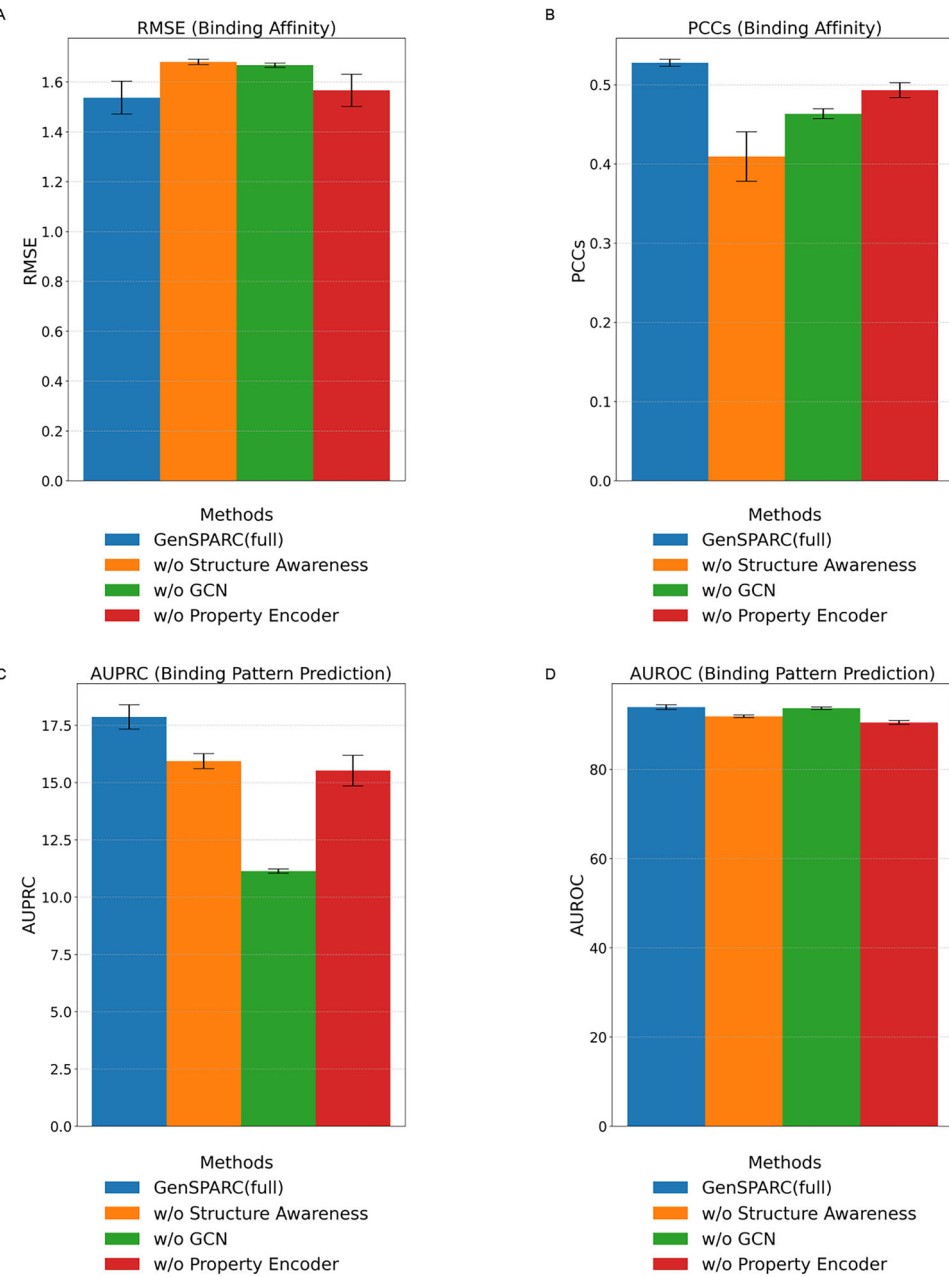

should address this issue by incorporating ensemble-based structural predictions of IDPs[60], allowing GenSPARC to better handle proteins with intrinsic disorder and extend its applicability to a wider range of biological contexts.

## Methods
### Datasets
**Original Karimi dataset**. The binding affinity and contact-map prediction experiments were conducted primarily on a publicly available dataset known as Cross-Interaction[11,20], which includes 4446 compound–protein pairs between 1287 proteins and 3672 compounds collected from PDBbind[61] and BindingDB[62]. To assess the model's generalizability, test data were split into four subsets as described for PSC–CPI and Cross–Interaction, based on whether the proteins or compounds were part of the training set: (1) Seen–Both (591 pairs), in which both the compound and protein had been seen; (2) Unseen–Comp (521 pairs), in which only the protein had been seen; (3) Unseen–Prot

(795 pairs), in which only the compound had been seen; and (4) Unseen–Both (202 pairs), in which neither had been seen.

To better evaluate the generalizability of the model, we introduced splits based on the similarity between proteins and compounds. First, we combined the four test subsets above into a single test set. For compounds, we calculated Tanimoto similarity between test and training compounds. For proteins, we considered both sequence and structural similarities. The data splitting process is illustrated in Fig. S1, while a detailed description is provided in Supplementary Methods. Fig. S2 shows the distribution of protein and compound similarity scores.

**Davis, KIBA, and Metz datasets**. Three widely recognized CPI benchmark datasets were employed to assess the efficacy of binding-affinity prediction: Davis[27], KIBA[28], and Metz[29]. The Davis dataset, which focused on interactions between 68 kinase inhibitors and 442 kinases, contained dissociation constant (Kd) values. These values were log-transformed to obtain pKd and enhance compatibility with

computational models. The KIBA dataset integrates various bioactivity measurements, including Kd, inhibition constant (Ki), and half-maximal inhibitory concentration (IC$_{50}$). The Metz dataset offers a detailed analysis of kinase inhibitor screening across 170 protein kinases. It employs rigorous statistical methods to ensure the quality and reliability of interaction data, making it a valuable resource for studying kinase-related interactions.

**DUD-E dataset**. We applied our method to a classic virtual screening task using the DUD-E[63] dataset and selected existing DL-based methods and docking programs for comparison. The DUD-E dataset comprised 102 targets from various protein families. Each target provided a set containing an average of 224 active (positive examples) and 10,000 decoy ligands (negative examples). The decoys were computationally selected to be physically similar, but topologically different from the active ligands. This task enabled the assessment of model performance in virtual screening tasks, where distinguishing active compounds from decoys is crucial.

## Protein input and encoder (SaProt)

Following the structure-aware vocabulary of SaProt[34], we enhanced the protein inputs by incorporating both the amino acid sequence and 3D geometric features. Traditional protein language models such as ESM-2[56] use only the amino acid sequence as input and lack any structural context. Instead, SaProt enriches each amino acid with a structural token derived from FoldSeek, which encodes 3D interactions at the residue level. This addition introduces crucial structural information absent from conventional approaches.

Specifically, a protein's primary sequence is represented as $P = (r_1, r_2, \ldots, r_n)$, where $r_i \in V$ corresponds to the residue at position $i$ in the amino acid alphabet $V$. To incorporate a 3D structure, we utilized a structure alphabet $S$, assigning each residue a structure token $s_j \in S$. This transformed $P = (r_1 s_1, r_2 s_2, \ldots, r_n s_n)$ into a structure-aware sequence. The alphabet $S$ contained 20 structure tokens plus an "unknown" token ("#"), which denoted missing structural information and enabled a more comprehensive representation of the protein's tertiary structure.

After constructing the structure-aware sequence, SaProt 650 M, which was trained on AlphaFold2-predicted structures, was employed to extract the embeddings. SaProt retains the architecture and parameter size of 650 M ESM-2 but expands the embedding layer with 441 structure-aware tokens, which replace the original 20 residue tokens. For each structure-aware sequence, SaProt generated embeddings by taking the last hidden state of its embedding layer (1280 dimensions). These embeddings were then projected into a lower-dimensional hidden space through a linear layer, $F_{prot}$, to integrate seamlessly with the compound embeddings in subsequent model stages.

This structure-aware enhancement provided a richer and more informative representation of proteins, addressing the limitations of sequence-only models such as ESM.

## Compound input and encoder

In the context of molecular compounds, we employed RDKit to convert the SMILES strings into molecular graphs $G_C = (V_C, E_C)$, where each atom $V_C$ was represented as a node with a feature vector. The vector included key atomic properties, such as the atomic symbol, aromaticity, Gasteiger charge, and additional features, as listed in Table S7. Further details on transforming the SMILES strings into molecular graphs can be found in the Supplementary Information.

The compound encoder is composed of three main elements: a molecular graph encoder, a molecular property encoder, and a fusion module that incorporates cross-attention and self-attention mechanisms.

The molecular graph encoder processes the molecular graph $G_C = (V_C, E_C)$ and learns $D$-dimensional node representation for each atom. As

a compound encoder, we employed GCNs, which are powerful variants of graph neural networks and are widely used for feature extraction from graph data. Given a molecular graph $G_C = (V_C, E_C)$, GCNs take their adjacency matrix $\mathcal{A}_C$ and node features $\mathcal{X}_C$ as inputs, and output a representation for each node. In our model, we used the following 3-layer GCN to process the molecular graph:

$$Z_{comp} = \hat{\mathcal{A}}\sigma\left(\hat{\mathcal{A}}\sigma\left(\hat{\mathcal{A}}\mathcal{X}_C W^0\right)W^1\right)W^2$$

where $\sigma = \text{ReLU}(\cdot)$ is the ReLU activation function. $\hat{\mathcal{A}} = D^{-\frac{1}{2}}(\mathcal{A}_C + I)D^{-\frac{1}{2}}$ represents the normalized adjacency matrix, where $\mathcal{A}_C$ is the adjacency matrix of the molecular graph, $I$ the identity matrix, and $\hat{D}$ the diagonal degree matrix for $(\mathcal{A}_C + I)$. $W^0 \in R^{d \times D}$, $W^1 \in R^{D \times D}$, and $W^2 \in R^{D \times D}$ are the learnable parameter matrices, with $D$ representing the hidden dimension size.

The molecular property encoder was adopted directly from the pre-trained property encoder in SPMM[35]. It processed 53 real-valued molecular properties (Table S4) and generated a 768-dimensional feature vector, denoted as $Z_{prop}$ (Supplementary Methods).

Finally, in the fusion module, we combined the compound embedding $Z_{comp}$ and property embedding $Z_{prop}$ using a molecular fusion module (Supplementary Methods). In this setup, the property embedding $Z_{prop}$ was treated as a query, whereas compound embedding served as the key and value in the multi-head attention (MHA) mechanism. This strategy refines compound embeddings based on their relevant properties, making the compounds more aware of both structural and property information:

$$F_{comp} = MHA\left(Z_{prop}, Z_{comp}, Z_{comp}\right)$$

$$F_{comp} = MHA\left(F_{comp}, F_{comp}, F_{comp}\right)$$

## MAN

To capture fine-grained binding information between a drug and target, our GenSPARC model utilized a fusion module to learn token-level interactions between the compound and protein representations provided by their respective encoders. As illustrated in Fig. 1C, we introduced a fusion module, termed MAN, which was designed to integrate protein and compound information. This module employs two distinct sets of projection matrices used to generate the query, key, and value representations as follows:

$$Q_{comp} = F_{comp}W_q^{comp}, K_{comp} = F_{comp}W_k^{comp}, V_{comp} = F_{comp}W_v^{comp}$$

$$Q_{prot} = F_{prot}W_q^{prot}, K_{prot} = F_{prot}W_k^{prot}, V_{prot} = F_{prot}W_v^{prot}$$

where $F_{comp}$ and $F_{prot}$ represent compound and protein embeddings, respectively. The MAN combines MHA and cross-attention mechanisms to refine the representation of each residue (in the protein) and each atom (in the compound). The refined representations were computed as follows:

$$F_{prot}^* = \frac{1}{2}\left[MHA\left(Q_{prot}, K_{prot}, V_{prot}\right) + MHA\left(Q_{prot}, K_{comp}, V_{comp}\right)\right]$$

$$F_{comp}^* = \frac{1}{2}\left[MHA\left(Q_{comp}, K_{comp}, V_{comp}\right) + MHA\left(Q_{comp}, K_{prot}, V_{prot}\right)\right]$$

Next, we applied residual connections and layer normalization, followed by a feed-forward network and another Add & Norm layer to further

refine the representations. Finally, we used a pooling strategy for both $F_{comp}$ and $F_{prot}$ separately, followed by concatenation. The final representation was as follows:

$$F^*_{prot} = FFN\Big(LayerNorm\big(F_{prot} + F^*_{prot}\big)\Big);$$

$$F^*_{comp} = FFN\Big(LayerNorm\big(F_{comp} + F^*_{comp}\big)\Big);$$

$$F^*_{prot} = LayerNorm\big(F_{prot} + F^*_{prot}\big)$$

$$F^*_{comp} = LayerNorm\big(F_{comp} + F^*_{comp}\big)$$

$$F_{joint} = \text{Concat}\Big(\text{MeanPool}\big(F^*_{comp}\big), \text{MeanPool}\big(F^*_{prot}\big)\Big)$$

where the *MeanPool* operation aggregated token-level interactions into a fixed-size representation. This fused representation was then used for the final prediction task, whereby we calculated the binding affinity $y^{pred}$ for the CPI prediction using a multilayer perceptron (MLP) regressor. The MLP was trained using an MSE loss function. The predicted affinities were calculated as follows:

$$y^{pred} = MLP\Big(F_{joint}\Big)$$

### Pairwise interaction network

To train the model to predict the interaction pattern (i.e., contact map) between a protein and a compound, we envisioned a 2D matrix representing these interactions. Protein embedding $F_{prot}$ and compound embedding $F_{comp}$ were used as inputs and were transformed to predict interaction intensities between residues and atoms by calculating the inner product between the transformed protein and the compound representations. These interaction intensities were normalized to produce the final contact map, denoted as $P_{inter}[m, n]$, which represented the interaction between the $m-th$ residue and the $n-th$ atom.

This approach enabled the model to effectively capture interaction patterns and generate a detailed contact map. The predicted contact map was then used to calculate the loss by comparing it with the ground-truth contact map for visualization purposes. Interaction intensity was normalized as follows:

$$P_{inter}[m, n] = \frac{P^*[m, n]}{\sum_{i,j} P^*[i, j]}$$

where $P^* = \text{Sigmoid}\Big((F_{prot})(F_{comp})^T\Big)$.

### Training and evaluation

**CPI task**. The CPI prediction task comprises two key objectives: affinity prediction and interaction pattern prediction. Given the true binding affinity $y^{true}_i$ between the $i-th$ protein and compound pair, the loss function for CPI strength prediction is defined as

$$\mathcal{L}^{affn} = \frac{1}{N}\sum_{i=1}^{N}\Big|y^{true}_i - y^{pred}_i\Big|^2,$$

Similarly, if the true interaction pattern $P_{true}[i, j]$ between the $i-th$ residue and the $j-th$ atom is known, the loss function for the CPI pattern

prediction is defined as

$$\mathcal{L}^{inter} = \frac{1}{M \cdot N}\sum_{i=1}^{M}\sum_{j=1}^{N}||P_{true}[i, j] - P_{inter}[i, j]||^2_F$$
$$+ \beta\Big(||P_{inter}[i, j]||_{group} + ||P_{inter}[i, j]||_{fused} + ||P_{inter}[i, j]||_1\Big),$$

where $||P_{inter}[i, j]||_{group}$[64], $||P_{inter}[i, j]||_{fused}$[65], and $||P_{inter}[i, j]||_1$ are the three sparsity regularization techniques used to control the sparsity of the interaction contact map $P_{inter}[i, j]$, as proposed by Karimi[11]. We reported the AUPRC and AUROC for this task.

Rather than treating CPI affinity and interaction pattern prediction as two discrete and independently trained tasks, we investigated the possibility of concurrently training them as a multitask learning model. We achieved this by integrating the two loss functions, $\mathcal{L}^{affn}$ and $\mathcal{L}^{inter}$, into a unified loss function: $\mathcal{L}^{integrate} = \mathcal{L}^{affn} + \mathcal{L}^{inter}$. The resulting model, denoted as GenSPARC–MT, enabled the training and prediction of both intermolecular affinity and atom–residue contacts within a unified framework.

**Virtual screening task**. For virtual screening of the DUD-E dataset, we evaluated GenSPARC using two approaches: (1) zero-shot and (2) cross-validation. In the zero-shot setting, the PDBBind[66] 2019 dataset, which was similarly employed in DrugCLIP[46], was used to exclude the targets present in DUD-E from the training set. Given that the training set contained only positive protein-compound pairs, we employed the in-batch negative construction strategy from CLIP[67] to generate negative examples. This strategy randomly pairs each protein with a different compound to create negative examples based on a well-founded assumption[68]. For the cross-validation setting, we followed the 3DCNN[47], DrugVQA[49], and AttentionSiteDTI[50] protocols to compare our model with benchmark methods. Specifically, we fine-tuned our model on the DUD-E dataset, performed 3-fold cross-validation on the dataset and reported the average performance across the evaluation metrics. Each fold was split based on the target, with similar targets within the same fold. Random undersampling was applied to the decoys to balance the training set, while keeping the test sets unbalanced for evaluation. We evaluated performance using several metrics from Drug-CLIP, including AUROC, BEDROC, EF (at 0.5%, 1%, and 5%), and RE (at thresholds of 0.5%, 1%, 2%, and 5%). BEDROC assigns more importance to top-ranked results, whereas EF and RE are commonly used metrics in virtual screening. Details about these metrics are provided in Supplementary Methods.

### Data availability

Data to repeat our analyses are available via GitHub (https://github.com/pfnet-research) and Zenodo (https://zenodo.org/records/17303785).

### Code availability

Our code and model weights are available at https://github.com/pfnet-research/GenSPARC.

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

## Acknowledgements
The authors thank the members of Preferred Networks, Inc., particularly J. Zhang and J. Iwasawa, for their critical comments on this manuscript.

## Author contributions
Y.Z. and A.T. proposed the concept and theory. Y.Z. implemented coding and analyzed data with assistance from A.T. and M.T. Y.Z., R.I., and A.T. wrote the manuscript. A.T. supervised the research.

## Competing interests
The authors declare no competing interests.
