## [Transparent Peer Review file · Communications Chemistry]

Generalizable Compound Protein Interaction Prediction with a Model Incorporating Protein Structure Aware and Compound Property Aware Language Model Representations

Corresponding Author: Dr Atsuhiko Tomita

Version 0:

Reviewer comments:

Reviewer #1

(Remarks to the Author)

The paper claims to develop a compound-protein predicting (CPI) model achieving state-of-the-art performance in tasks of binding affinity prediction, contact prediction, and virtual screening.

* The title is misleading in terms of "Language Models". It turns my first impression to that the model is built on ChatGPT or sth similar. But it is actually protein/chemical language models which are only parts of the entire pipeline.

* For the binding affinity prediction and contact prediction benchmarking, the numbers are pretty impressive. For a more solid comparison, could you add one more competitor method of "MONN: A Multi-objective Neural Network for Predicting Compound-Protein Interactions and Affinities"?

* What is the "upper bound" of these benchmarking numbers, in terms of if I know the 3D features (can be predicted)? How far can the non-3D predictor approach them? Can you use your datasets to assess this latest model called Boltz which claims to accurately predict affinity with 3D features (<https://github.com/jwohlwend/boltz>)?

* For the protein-compound interaction patterns, for those two showcases proteins in Figure 2, how the top predicted ligands differ across the three methods? Can you analyze what new class of molecules are prioritized by your model?

* For the virtual screening evaluation, I am not super clear about the difference between the zero-shot setup and cross-validation: Do you do fine tuning in the latter but not the former? If that is the case, why are the compared methods different between Tables 6 and 7? For instance DrugCLIP is not reported in the cross-validation setting.

* For the virtual screening evaluation, the zero-shot performance is not good but cross-validation one is good. Does finetuning on seen targets + unseen ligands help more or unseen targets + seen ligands help more?

* Figure 3 needs to be re-made. The patterns are not super visually readable.

* Is the predictor you built ready and easy to be used by biologists? Can they just upload their protein sequences and SMILES into some website and return the prediction? Codes seem to be unavailable in the text.

Reviewer #2

(Remarks to the Author)

I co-reviewed this manuscript with one of the reviewers who provided the listed reports. This is part of the Communications Chemistry initiative to facilitate training in peer review and to provide appropriate recognition for Early Career Researchers who co-review manuscripts.

Reviewer #3

(Remarks to the Author)

The paper introduces GenSPARC, a model for compound–protein interaction (CPI) prediction that integrates structure-aware protein representations and property-aware compound representations. The model is evaluated on the Karimi dataset using four generalization splits (Seen-Both, Unseen-Compound, Unseen-Protein, Unseen-Both), and benchmarked against standard CPI datasets (DAVIS, KIBA, and Metz). An ablation study is conducted to assess the contribution of each model component. While the work claims to achieve SOTA performance, several concerns remain.

Major Concerns

Insufficient Context for Related Work. The paper claims to address the issue of inadequate integration of 3D protein structural information in current models (line 55). However, despite the existence of numerous 3D structure-aware models, only PSC-CPI is mentioned in the introduction. The authors should first clearly define the problem and provide a more comprehensive discussion of relevant structure-aware models, explaining how GenSPARC addresses the limitations of these existing approaches.

Incomplete Baseline Comparison Undermines SOTA Claims. The paper claims that GenSPARC outperforms existing state-of-the-art (SOTA) models. However, it does not include several recent and relevant structure-aware CPI prediction methods, such as SIGN, GraphscoreDTA, Folding-Docking-Affinity, etc. In the Karimi dataset benchmarks, only Cross-Interaction and PSC-CPI are included as baselines, which limits the strength of the performance claims. A similar issue exists in the DAVIS, KIBA, and Metz dataset evaluations, where key graph-based models are also excluded, further weakening the strong generalizability and SOTA performance claims.

Limited Novelty. The methodological novelty is limited. GenSPARC merely stacked existing pre-trained models without introducing fundamentally new architectural innovations or techniques. Additionally, it does not clearly tackle a new or underexplored problem in CPI prediction.

Inconsistent Generalizability Evaluation. While the Karimi dataset is evaluated under four generalization settings, the same level of analysis is not extended to DAVIS, KIBA, or Metz datasets, which are limited to Unseen-Both split. This makes it difficult to validate the generalizability claims beyond the Karimi benchmark.

Lack of Motivation for Compound Property Encoder. The compound property encoder is shown (via ablation) to contribute to improved performance, but the paper fails to explain the rationale for including it. A clear justification is needed in the introduction to align with the experimental design.

Loss of 3D Information Not Quantified. The model uses FoldSeek to encode AlphaFold-generated protein structures into structure-aware sequences. However, the potential information loss during this transformation is neither quantified nor discussed.

Minor Suggestions

Title Clarity: The phrase “structure-aware and property-aware representations” may cause ambiguity. Consider revising to more clearly distinguish that “structure-aware” refers to proteins and “property-aware” to compounds.

Figure 1: Clarify which components are trainable and which are frozen. A visual distinction or annotation would improve interpretability.

Table 1: Report the number of folds used in cross-validation and include standard deviations to assess result stability.

Figure 2: Add a legend to explain the color coding used in the visualization.

Lines 214–218: The optimistic summary should be tempered by acknowledging the performance drop observed when using AlphaFold structures, which highlights a limitation of the approach.

Reviewer #4

(Remarks to the Author)

GenSPARC is a new deep learning model for predicting contact maps between proteins and compounds. Existing models are used as protein and compound encoders. Its performance is good but I have the following reservations.

1. This work seems to claim that it proposes a new representation (R in GenSPARC). What does it mean, when both protein and compounds are represented via existing encoders.
2. GenSPARC is shown to be better in accuracy in comparison to existing models. However, as shown in Figure 2, it still misses a lot of contacts. Is there any hope that this kind of deep learning models can be improved to the level of molecular dynamics? Please discuss.
3. Do the datasets used incorporate intrinsically disordered proteins? If yes, please show some results about them. If not, please discuss about the prospects about how to deal with such proteins.

Version 1:

Reviewer comments:

Reviewer #1

(Remarks to the Author)

The authors have answer all my questions. I do not have further comments to the manuscript.

Reviewer #2

(Remarks to the Author)

I co-reviewed this manuscript with one of the reviewers who provided the listed reports. This is part of the Communications Chemistry initiative to facilitate training in peer review and to provide appropriate recognition for Early Career Researchers who co-review manuscripts.

Reviewer #3

(Remarks to the Author)

The authors had made a substantial revision. We still have the following two comments:

Original comment: "Insufficient Context for Related Work. The paper claims to address the issue of inadequate integration of 3D protein structural information in current models (line 55). However, despite the existence of numerous 3D structure-aware models, only PSC-CPI is mentioned in the introduction. The authors should first clearly define the problem and provide a more comprehensive discussion of relevant structure-aware models, explaining how GenSPARC addresses the limitations of these existing approaches."

Authors' response:

We thank the reviewer for this valuable suggestion.

In the revised manuscript, we have clarified that our main focus is on CPI prediction from protein sequences and compound SMILES. We also recognize the importance of structure-aware models which predict binding affinity between proteins and compounds using experimentally determined complex structures or binding pocket information. Accordingly, we have expanded the Introduction to include several representative structures-aware DTA models, highlighting their contrast with CPI tasks and the associated challenges.

Reviewer's additional comment:

My primary concern is that the authors' revision introduces an artificial distinction between "Compound-Protein Interaction (CPI) prediction" and "Drug Target Affinity (DTA)" models. But, in my point of view, these refer to the same task: predicting interaction strength between a compound (drug) and a protein (target). In addition, the authors now seem to define "structure-based DTA" exclusively as models that are "docking-based"—that is, models that require a protein-ligand complex structure or pose as input. But structure-based DTA models are not limited to docking-based, they could be docking-free methods, like the authors' work. My concern was, and remains, the lack of discussion regarding 3D structure-aware models that do not require a docking pose (docking-free) .

Furthermore, the new text added by the authors contains specific claims that are inaccurate or unsubstantiated.

The claim "Recently, Fold-Docking-Affinity extended a structure-based DTA model by combining AlphaFold2 and DiffDock to predict affinities without relying on experimental structures. However, its application results have been limited to kinase targets" is a misinterpretation. The authors of that paper used kinase datasets for validation of their pipeline; this does not mean the method itself is limited to that target class.

The final sentence ("...and the combination of AlphaFold2 structures with DiffDock has shown limited success across diverse targets, indicating that extending DTA models to CPI tasks remains difficult") is vague and poorly justified. How does the (alleged) difficulty of a docking-based approach (which they term DTA) logically support the conclusion that "extending DTA models to CPI tasks remains difficult," This connection is unclear and seems to obscure the problem rather than clarify it.

Original comment: "Incomplete Baseline Comparison Undermines SOTA Claims. The paper claims that GenSPARC outperforms existing state-of-the-art (SOTA) models. However, it does not include several recent and relevant structure-aware CPI prediction methods, such as SIGN, GraphScoreDTA, Folding-Docking-Affinity, etc. In the Karimi dataset benchmarks, only Cross-Interaction and PSC-CPI are included as baselines, which limits the strength of the performance claims. A similar issue exists in the DAVIS, KIBA, and Metz dataset evaluations, where key graph-based models are also excluded, further weakening the strong generalizability and SOTA performance claims."

Authors' response:

We appreciate the reviewer's insightful comment regarding the baseline selection. We first note that several of the mentioned structure-aware CPI methods (e.g., SIGN, GraphScoreDTA, and Fold-Docking-Affinity) are currently difficult to reproduce or execute, as their publicly available implementations are incomplete or restricted.

- SIGN: the publicly available information on the SIGN pipeline is limited (<https://github.com/agave233/SIGN/issues/4>), and the software required for the preprocessing of inputs is not accessible to us due to license restrictions.

- GraphScoreDTA: the code to generate the feature map file is not provided (<https://github.com/KailiWang1/GraphscoreDTA/issues/1>).

- Fold-Docking-Affinity is indeed a valuable method, as it enables the prediction and evaluation of complex structures following well-established protocols. However, the authors themselves noted in the official repository that the provided code is not practically executable (<https://github.com/ZhiGroup/FDA>). Despite our best efforts, we found it extremely challenging not only to preprocess and train on new datasets, but even to perform inference on them.

Even with available implementations, fair comparison would remain difficult because many of these methods require experimentally determined complex structures or pre-defined binding pockets as input. In contrast, our task is strictly defined as CPI prediction from protein sequences and compound SMILES only, without assuming prior structural knowledge. Furthermore, the Karimi dataset focuses on binding pattern prediction, which cannot be directly evaluated by models that rely on pre-formed complexes. Given these limitations and in an effort to ensure the highest possible quality of our benchmarks, we adopted GraphBAN (<https://www.nature.com/articles/s41467-025-57536-9>), a recently published method in Nature Communications that demonstrated strong performance and is reproducible.

Additionally, following suggestions from other reviewers, we included MONN

(<https://www.sciencedirect.com/science/article/pii/S2405471220300818>) as an additional baseline in our comparison experiments. We modified and included MONN and GraphBAN as representative and competitive baselines in our comparison experiments (Tables 1-5). We note that while MONN only requires protein sequences and compound SMILES during inference, it relies on interaction information during training; consequently, it was not evaluated on benchmark datasets such as KIBA, Davis, or Metz). In addition, we clarified that, for these datasets, only methods not requiring complex structures were considered, in order to ensure a fair comparison, thereby toning down the original statement.

Reviewer's additional comment:

I thank the authors for their response and for including two additional baselines, GraphBAN and MONN.

However, the rebuttal does not resolve my primary concern regarding the incomplete baseline comparison. This is essential for substantiating the paper's state-of-the-art (SOTA) claims, and the justifications for excluding key docking-based structure-aware models are not convincing.

1. On the Reproducibility of Baselines

The claim that reproducing Fold-Docking-Affinity (FDA) is "extremely challenging" or "not practically executable" is inaccurate. The FDA pipeline is a composition of well-known, independently runnable, and publicly available tools: ColabFold (for structure prediction), DiffDock (for docking), and a GNN-based affinity predictor (like GIGN).

Each of these components is functional and widely used. While the authors' specific repository may be for "reference purposes only," the methodology itself is clearly defined and reproducible. Similarly, for Boltz-2, even if training the full structural module is infeasible, it is certainly feasible to retrain their affinity module using the authors' datasets. This would provide a crucial reference point.

2. On the "Task Definition"

The authors' main defense is to state their "task is strictly defined as CPI prediction from protein sequences and compound SMILES only." This definition is precisely the problem. It is an artificial constraint that allows the paper to ignore the most significant recent advances in the field. The current SOTA for this exact problem (predicting affinity from protein sequence and SMILES) involves generating binding structures as an intermediate step.

My original point, which Reviewer #1 also raised, is that a SOTA claim is meaningless if it does not compare against methods that use these computed structures. It is better to know how GenSPARC performs against a baseline that first computes binding poses (e.g., via Boltz-2 or an FDA-like pipeline) and then predicts affinity.

Claiming this is a "different task" is a weak defense. It is the same input-output problem; these docking-based pipelines simply represent the true SOTA competitors. I appreciate that contact-map prediction can be tricky (e.g., with data-leakage risks). Even so, for the affinity-prediction benchmarks, including these baselines seems important. To support the SOTA claims, I'd encourage adding a comparison against models that leverage computed binding (such as Boltz-2 or FDA) for the affinity tasks.

Reviewer #4

(Remarks to the Author)

The authors addressed all of my comments appropriately.

Reviewer #1

“The paper claims to develop a compound-protein predicting (CPI) model achieving state-of-the-art performance in tasks of binding affinity prediction, contact prediction, and virtual screening.”

We appreciate Reviewer #1’s time and effort, as well as the constructive and encouraging comments regarding our manuscript. The specific points mentioned by Reviewer #1 are addressed below.

“* The title is misleading in terms of "Language Models". It turns my first impression to that the model is built on ChatGPT or sth similar. But it is actually protein/chemical language models which are only parts of the entire pipeline.”

We apologize for the confusion caused by the previous title. We revised the title to “Generalizable Compound–Protein Interaction Prediction with a Model Incorporating Protein Structure-Aware and Compound Property-Aware Language Model Representations” and updated the model abbreviation to “GenSPARC: A Model with **Generalized Structure-** and **Property-Aware Representations** of Protein and Chemical Language Models for Compound–Protein Interaction Prediction”

“* For the binding affinity prediction and contact prediction benchmarking, the numbers are pretty impressive. For a more solid comparison, could you add one more competitor method of "MONN: A Multi-objective Neural Network for Predicting Compound-Protein Interactions and Affinities"?”

We thank the reviewer for this valuable suggestion. We have added MONN as a competitor method and included its performance on the Karimi dataset in our revised benchmarking results (Table 1-4). We note that MONN's training requires interaction information between compounds and proteins. While this information is available in the Karimi dataset, it is not present in the KIBA, Davis, and Mets datasets. Therefore, we have included MONN comparisons only for the Karimi dataset. Our results show that GenSPARC achieves competitive performance compared to MONN while demonstrating broader applicability across datasets with varying levels of available information.

“* What is the "upper bound" of these benchmarking numbers, in terms of if I know the 3D features (can be predicted)? How far can the non-3D predictor approach them? Can you use your datasets to assess this latest model called Boltz which claims to accurately predict affinity with 3D features (<https://github.com/jwohlwend/boltz>)?”

We appreciate this comment. It is challenging to directly compare our method with Boltz on the dataset used in this study, because Boltz has already been trained on the structures and affinity data included in the CPI benchmark datasets. Therefore, reproducing the Seen/Unseen split of CPI benchmark datasets used in this paper, which is very important for a fair comparison, is impossible without the retraining of Boltz from scratch.

However, as reported in Table 6, state-of-the-art structure-aware model such as DrugCLIP-PDB achieve high performance when accurate 3D structural information (experimental structures) is available. Importantly, our approach can achieve high performance even in scenarios where precise 3D structural knowledge is not available, by leveraging predicted structures such as those from AlphaFold2, thereby providing a practical alternative to existing methods.

“* For the protein-compound interaction patterns, for those two showcases proteins in Figure 2, how the top predicted ligands differ across the three methods? Can you analyze what new class of molecules are prioritized by your model?”

We thank the reviewer for this thoughtful question. We performed a comprehensive analysis on the

Karimi dataset, examining the top predicted ligands across the three methods and evaluating correlations between model rankings. In the figure below, we selected the top 50 compounds with the highest AUPRC values using each method—CrossInteraction, PSC-CPI, and GenSPARC—and then visualized those compounds where GenSPARC demonstrates superior prediction performance compared to both CrossInteraction and PSC-CPI. This analysis suggests that there are no clear preferences for specific classes of molecules by our model.

Figure: Molecules which have higher AUPRC values in GenSPARC prediction than CrossInteraction (upper panel) and PSC-CPI (lower panel).

To capture the overall tendencies of these top 50 molecules of each prediction method, we mapped the molecules in the 2D chemical space. For each of the three models—CrossInteraction, PSC-CPI, and GenSPARC—we selected the top 50 molecules with the highest PRC-AUC values, computed ECFP fingerprints (radius=2, nbit=2048), and projected them onto a two-dimensional PCA plane (contribution of each PC is indicated in the axis label). The points from these methods largely overlapped, with no clear tendencies or distinct clusters in the chemical space for each method. This

observation suggests that none of these three methods exhibits a preference for particular areas of chemical space. Overall, our model shows uniformly high accuracy across diverse compound types, indicating improved robustness rather than selective enrichment of a particular molecular category.

“* For the virtual screening evaluation, I am not super clear about the difference between the zero-shot setup and cross-validation: Do you do fine tuning in the latter but not the former? If that is the case, why are the compared methods different between Tables 6 and 7? For instance DrugCLIP is not reported in the cross-validation setting.”

We thank the reviewer for the opportunity to clarify this point. In the zero-shot setting, our model is trained once on PDBBind 2019 (excluding DUD-E targets) and directly evaluated on DUD-E without further tuning, assessing generalization to unseen targets. In contrast, in the cross-validation setting, we fine-tune on DUD-E following the three-fold target-level split, which enables fair comparison with prior works such as 3DCNN, DrugVQA, and AttentionSiteDTI. In the revised manuscript, we have also added DrugCLIP cross-validation results, updated the corresponding table (Table 7), and revised the main text and Methods section to clearly distinguish between the two evaluation settings (Method, p21).

“* For the virtual screening evaluation, the zero-shot performance is not good but cross-validation one is good. Does finetuning on seen targets + unseen ligands help more or unseen targets + seen ligands help more?”

We appreciate the reviewer’s thoughtful question. As we cannot strictly control the Seen/Unseen splits in the same way across datasets, it is difficult to quantitatively disentangle the relative contributions of “Seen targets + Unseen ligands” versus “Unseen targets + Seen ligands.” Nevertheless, in general, affinity prediction tasks tend to show lower accuracy on unseen proteins compared with unseen compounds (Table 4). This suggests that fine-tuning with Seen targets + Unseen ligands is likely to provide greater benefit, as the model can leverage prior knowledge of the target while adapting to new ligands.

“* Figure 3 needs to be re-made. The patterns are not super visually readable.”

We thank the reviewer for this helpful suggestion. To improve readability, we have revised Figure 3 by extracting the most relevant patterns and presenting them in a clearer format. The full set of detailed graphs has been moved to the Supplementary Information (Figure S3).

“* Is the predictor you built ready and easy to be used by biologists? Can they just upload their protein sequences and SMILES into some website and return the prediction? Codes seem to be unavailable in the text.”

We apologize for the unavailability of the code in the original submission. We have now uploaded the code to GitHub: [<https://github.com/pfnet-research/GenSPARC>]. Although we do not plan to host a web server, all code, trained models, and datasets will be publicly accessible, with detailed documentation to enable anyone to use the predictor locally.

Reviewer #2

“I co-reviewed this manuscript with one of the reviewers who provided the listed reports. This is part of the Communications Chemistry initiative to facilitate training in peer review and to provide appropriate recognition for Early Career Researchers who co-review manuscripts.”

We sincerely thank Reviewer #2 for taking the time to co-review this manuscript and for their collaboration with Reviewer #4. We greatly appreciate the constructive and encouraging feedback provided through this joint effort, which has been invaluable in improving our work.

Reviewer #3

“The paper introduces GenSPARC, a model for compound–protein interaction (CPI) prediction that integrates structure-aware protein representations and property-aware compound representations. The model is evaluated on the Karimi dataset using four generalization splits (Seen-Both, Unseen-Compound, Unseen-Protein, Unseen-Both), and benchmarked against standard CPI datasets (DAVIS, KIBA, and Metz). An ablation study is conducted to assess the contribution of each model component. While the work claims to achieve SOTA performance, several concerns remain.”

We appreciate Reviewer #3’s time and effort, as well as the constructive and encouraging comments regarding our manuscript. The specific points mentioned by Reviewer #3 are addressed below.

Major Concerns

“Insufficient Context for Related Work. The paper claims to address the issue of inadequate integration of 3D protein structural information in current models (line 55). However, despite the existence of numerous 3D structure-aware models, only PSC-CPI is mentioned in the introduction. The authors should first clearly define the problem and provide a more comprehensive discussion of relevant structure-aware models, explaining how GenSPARC addresses the limitations of these existing approaches.”

We thank the reviewer for this valuable suggestion.

In the revised manuscript, we have clarified that our main focus is on CPI prediction from protein sequences and compound SMILES. We also recognize the importance of structure-aware models which predict binding affinity between proteins and compounds using experimentally determined complex structures or binding pocket information. Accordingly, we have expanded the Introduction to include several representative structure-aware DTA models, highlighting their contrast with CPI tasks and the associated challenges.

“Incomplete Baseline Comparison Undermines SOTA Claims. The paper claims that GenSPARC outperforms existing state-of-the-art (SOTA) models. However, it does not include several recent and relevant structure-aware CPI prediction methods, such as SIGN, GraphScoreDTA, Folding-Docking-Affinity, etc. In the Karimi dataset benchmarks, only Cross-Interaction and PSC-CPI are included as baselines, which limits the strength of the performance claims. A similar issue exists in the DAVIS, KIBA, and Metz dataset evaluations, where key graph-based models are also excluded, further weakening the strong generalizability and SOTA performance claims.”

We appreciate the reviewer’s insightful comment regarding the baseline selection. We first note that several of the mentioned structure-aware CPI methods (e.g., SIGN, GraphScoreDTA, and Fold-Docking-Affinity) are currently difficult to reproduce or execute, as their publicly available implementations are incomplete or restricted.

- SIGN: the publicly available information on the SIGN pipeline is limited (<https://github.com/agave233/SIGN/issues/4>), and the software required for the preprocessing of inputs is not accessible to us due to license restrictions.
- GraphScoreDTA: the code to generate the feature map file is not provide (<https://github.com/KailiWang1/GraphscoreDTA/issues/1>).
- Fold-Docking-Affinity is indeed a valuable method, as it enables the prediction and evaluation of complex structures following well-established protocols. However, the authors themselves noted in the official repository that the provided code is not practically executable (<https://github.com/ZhiGroup/FDA>). Despite our best efforts, we found it extremely challenging not only to preprocess and train on new datasets, but even to perform inference on

them.

Even with available implementations, fair comparison would remain difficult because many of these methods require experimentally determined complex structures or pre-defined binding pockets as input. In contrast, our task is strictly defined as CPI prediction from protein sequences and compound SMILES only, without assuming prior structural knowledge. Furthermore, the Karimi dataset focuses on binding pattern prediction, which cannot be directly evaluated by models that rely on pre-formed complexes.

Given these limitations and in an effort to ensure the highest possible quality of our benchmarks, we adopted GraphBAN (<https://www.nature.com/articles/s41467-025-57536-9>), a recently published method in Nature Communications that demonstrated strong performance and is reproducible. Additionally, following suggestions from other reviewers, we included MONN (<https://www.sciencedirect.com/science/article/pii/S2405471220300818>) as an additional baseline in our comparison experiments. We modified and included MONN and GraphBAN as representative and competitive baselines in our comparison experiments (Tables 1-5, We note that while MONN only requires protein sequences and compound SMILES during inference, it relies on interaction information during training; consequently, it was not evaluated on benchmark datasets such as KIBA, Davis, or Metz). In addition, we clarified that, for these datasets, only methods not requiring complex structures were considered, in order to ensure a fair comparison, thereby toning down the original statement.

“Limited Novelty. The methodological novelty is limited. GenSPARC merely stacked existing pre-trained models without introducing fundamentally new architectural innovations or techniques. Additionally, it does not clearly tackle a new or underexplored problem in CPI prediction.”

We appreciate the reviewer’s concern regarding methodological novelty. We would like to clarify that the use of pre-trained encoders such as ESM-2 for proteins and SELFormer for compounds is now standard practice in the CPI prediction field, and the value of a framework lies largely in how these representations are integrated and adapted. In GenSPARC, we made several original contributions beyond simply stacking models: (i) designing a compound encoder that integrates information from a pretrained SPM property encoder and our custom molecular graph encoder based on GCNs, (ii) developing a multimodal integration strategy for protein and compound features, and (iii) establishing rigorous dataset splits to assess generalizability.

We respectfully believe that these contributions provide sufficient novelty and practical significance within the current research landscape, and we hope the reviewer may view our work in this context.

“Inconsistent Generalizability Evaluation. While the Karimi dataset is evaluated under four generalization settings, the same level of analysis is not extended to DAVIS, KIBA, or Metz datasets, which are limited to Unseen-Both split. This makes it difficult to validate the generalizability claims beyond the Karimi benchmark.”

We thank the reviewer for this important comment and apologize for initially showing only the Unseen-Both split, which was intended to focus on the most challenging and interesting cases. In response to this comment, we have added additional splits of the DAVIS, KIBA, and Metz datasets (Seen-Both, Unseen-Protein, and Unseen-Compound) in the Supplementary Information (Table S5-7). This allows for consistent evaluation across all datasets and facilitates a more thorough validation of the model’s generalizability. Across these additional evaluations, GenSPARC exhibited consistently

superior performance in the most unseen test settings, demonstrating strong generalization capability beyond the training distribution.

“Lack of Motivation for Compound Property Encoder. The compound property encoder is shown (via ablation) to contribute to improved performance, but the paper fails to explain the rationale for including it. A clear justification is needed in the introduction to align with the experimental design.”

We thank the reviewer for this insightful comment. The motivation for including the compound property encoder can be summarized in two points.

First, in the original work on SPMM, it was reported that incorporating molecular property features led to consistent improvements across downstream tasks. Building on this observation, we integrated a compound property encoder into GenSPARC to enrich compound representations with complementary information beyond the molecular graph.

Second, as clarified in our earlier response, a central goal of GenSPARC is to improve generalizability by leveraging multimodal information. Incorporating the compound property encoder allows us to capture diverse molecular features in addition to sequence and structural information, which aligns directly with this motivation. Indeed, our ablation study demonstrates that the compound property encoder consistently contributes to improved generalization performance across benchmarks.

We have revised the introduction to more clearly justify this design choice and to align it with the overall experimental motivation of GenSPARC.

As a supplementary analysis, we evaluated the contribution of each property dimension by performing an ablation study in which each of the 53 dimensions was individually masked during training and inference. While masking had minimal effect on the Seen both (=Test) set, some dimensions showed noticeable impact on the Pearson correlation in the Unseen both setting. Ranking the properties by the largest decrease in Pearson correlation under this condition yields:

- Drop ID 19, Prop: HallKierAlpha, Pearson: 0.3641
- Drop ID 32, Prop: MolWt, Pearson: 0.3885
- Drop ID 27, Prop: MaxEStateIndex, Pearson: 0.3979
- Drop ID 52, Prop: QED, Pearson: 0.3985
- Drop ID 29, Prop: MinEStateIndex, Pearson: 0.4250

These include descriptors reflecting molecular shape (e.g., HallKierAlpha), overall molecular properties (e.g., MolWt, QED), and atom-level electronic states (e.g., EStateIndex). This suggests that, particularly when both protein and compound are unseen, these descriptors contribute to enhancing model performance even in the absence of corresponding training data.

Metrics - Test

Metrics - Unseen Comp Original

Metrics - Unseen Prot

Metrics - Unseen Both

“Loss of 3D Information Not Quantified. The model uses FoldSeek to encode AlphaFold-generated protein structures into structure-aware sequences. However, the potential information loss during this transformation is neither quantified nor discussed.”

We thank the reviewer for this comment. Directly quantifying the potential information loss during the transformation to FoldSeek’s 3Di representation is challenging; accordingly, we note this as a limitation in the Discussion section, acknowledging that some fine-grained structural information may be lost. We also emphasize that developing approaches to further reduce structural information loss without compromising generalization represents an important direction for future work.

Minor Suggestions

“Title Clarity: The phrase “structure-aware and property-aware representations” may cause ambiguity. Consider revising to more clearly distinguish that “structure-aware” refers to proteins and “property-aware” to compounds.”

We thank the reviewer for this suggestion. We revised the title to “Generalizable Compound–Protein Interaction Prediction with a Model Incorporating Protein Structure-Aware and Compound Property-Aware Language Model Representations”.

“Figure 1: Clarify which components are trainable and which are frozen. A visual distinction or annotation would improve interpretability.”

Thank you for the suggestion. We have added a frozen mark in Fig.1B to indicate the encoder components that are pretrained and kept fixed, in order to clarify which parts are frozen versus trainable.

“Table 1: Report the number of folds used in cross-validation and include standard deviations to assess result stability.”

Thank you for the comment. In line with prior work on the Karimi dataset (e.g., Cross-Interaction, PSC-CPI), we did not perform k -fold cross-validation. Thus, standard deviations are not applicable.

“Figure 2: Add a legend to explain the color coding used in the visualization.”

We appreciate the reviewer's feedback. In the revised version of Figure 2, we explicitly added a legend/note clarifying that the color coding reflects the probability of interaction.

“Lines 214–218: The optimistic summary should be tempered by acknowledging the performance drop observed when using AlphaFold structures, which highlights a limitation of the approach.”

We appreciate the reviewer's suggestion. We modified the manuscript to tone down the statement (P11) and now describe the findings based on what can be quantitatively concluded from the results, including the performance drop observed when using AlphaFold structures.

Reviewer #4

“GenSPARC is a new deep learning model for predicting contact maps between proteins and compounds. Existing models are used as protein and compound encoders. Its performance is good but I have the following reservations.”

We appreciate Reviewer #4's time and effort, as well as the constructive and encouraging comments regarding our manuscript. The specific points mentioned by Reviewer #4 are addressed below.

“1. This work seems to claim that it proposes a new representation (R in GenSPARC). What does it mean, when both protein and compounds are represented via existing encoders.”

We thank the reviewer for raising this insightful comment regarding the novelty of the representations used in GenSPARC. We would like to clarify that the use of pre-trained encoders, such as ESM-2 for proteins and SELFormer for compounds, is now standard practice in the CPI prediction field. The novelty in terms of “representation” in our work refers not to the individual encoders themselves, but to how these representations are combined and integrated within our framework.

Within this combined representation, we have introduced several original contributions beyond simply stacking pre-trained models:

- (i) designing a compound property encoder based on GNNs,
- (ii) developing a multimodal integration strategy for protein and compound features, and
- (iii) establishing rigorous dataset splits to assess generalizability.

To prevent potential misinterpretation from the original title, we have also revised it to:

“Generalizable Compound–Protein Interaction Prediction with a Model Incorporating Protein Structure-Aware and Compound Property-Aware Language Model Representations.”

We respectfully believe that these contributions provide sufficient novelty and practical significance within the current research landscape, and we hope the reviewer may view our work in this context.

“2. GenSPARC is shown to be better in accuracy in comparison to existing models. However, as shown in Figure 2, it still misses a lot of contacts. Is there any hope that this kind of deep learning models can be improved to the level of molecular dynamics? Please discuss.”

We appreciate the reviewer for this comment. We have added a note in the Discussion section (P14).

While GenSPARC improves accuracy, some contacts are still missed. A potential strategy is to use predicted interactions as constraints in complex structure prediction models such as AlphaFold3 or Boltz, enabling the generation of plausible complex structures consistent with GenSPARC's predictions, which can then be further refined to the level of molecular dynamics.

“3. Do the datasets used incorporate intrinsically disordered proteins? If yes, please show some results about them. If not, please discuss about the prospects about how to deal with such proteins.”

We thank the reviewer for this comment. The datasets used in our study do not include intrinsically disordered proteins (IDPs). Therefore, we cannot claim any advantage of our approach over models based solely on sequence encoders such as ESM when it comes to IDPs. We have added a note (P14) in the revised manuscript clarifying this point and briefly discuss potential future directions for handling IDPs, such as integrating disorder prediction or flexible representations into the framework.

Preferred Networks Inc.

1-6-1 Otemachi, Chiyoda-ku, Tokyo 100-0004, Japan

November 30, 2025

Dr Huijuan Guo
Senior Editor,
Communications Chemistry,

Dear Dr Huijuan Guo,

COMMSCHEM-25-0384B

Thank you so much for your positive reply on November 13th with respect to our manuscript. We would now like to submit the revised manuscript, entitled “*Generalizable Compound Protein Interaction Prediction with a Model Incorporating Protein Structure Aware and Compound Property Aware Language Model Representations*” by Zhang, et al., for consideration as a research article in *Communications Chemistry*.

We greatly appreciate the constructive and encouraging comments from both reviewers. In response to the remaining concerns raised by Reviewer #3, we have included in the Discussion section a consideration of potential future studies, specifically discussing the Boltz-2 framework that consistently integrates structure prediction and binding affinity prediction. Please find below our point-by-point responses addressing the remaining concerns of Reviewer #3. We hope that these responses may serve as a useful reference within the scope of the editor’s review. Furthermore, we have ensured that the manuscript complies with the journal’s policies and formatting guidelines.

All requested editorial revisions have been addressed, and we have uploaded the completed table outlining the changes made. We believe that the current version of the manuscript reflects a thorough and careful revision, and we hope it meets the requirements for publication.

We sincerely appreciate your continued support and guidance, and we look forward to your consideration of this revised manuscript.

Thank you very much for your time and attention.
Sincerely,

Atsuhiko Tomita

Reviewer #3

“My primary concern is that the authors' revision introduces an artificial distinction between "Compound-Protein Interaction (CPI) prediction" and "Drug Target Affinity (DTA)" models. But, in my point of view, these refer to the same task: predicting interaction strength between a compound (drug) and a protein (target). In addition, the authors now seem to define "structure-based DTA" exclusively as models that are "docking-based"—that is, models that require a protein-ligand complex structure or pose as input. But structure-based DTA models are not limited to docking-based, they could be docking-free methods, like the authors' work. My concern was, and remains, the lack of discussion regarding 3D structure-aware models that do not require a docking pose (docking-free).

Furthermore, the new text added by the authors contains specific claims that are inaccurate or unsubstantiated.

The claim "Recently, Fold-Docking-Affinity extended a structure-based DTA model by combining AlphaFold2 and DiffDock to predict affinities without relying on experimental structures. However, its application results have been limited to kinase targets" is a misinterpretation. The authors of that paper used kinase datasets for validation of their pipeline; this does not mean the method itself is limited to that target class.

The final sentence ("...and the combination of AlphaFold2 structures with DiffDock has shown limited success across diverse targets, indicating that extending DTA models to CPI tasks remains difficult") is vague and poorly justified. How does the (alleged) difficulty of a docking-based approach (which they term DTA) logically support the conclusion that "extending DTA models to CPI tasks remains difficult," This connection is unclear and seems to obscure the problem rather than clarify it.”

We thank the reviewer for the detailed comments. We agree that, in principle, Fold-Docking-Affinity (FDA) can be applied to any target. However, in the original publication, the authors clearly demonstrate results only for kinase datasets and explicitly mention that applying the method to other protein families remains a challenge.

Moreover, predicting protein-ligand complex structures using AlphaFold2 and DiffDock is known to be challenging as described in the main text. In this context, it is difficult to expect that a graph-based framework in the FDA could accurately predict binding affinities from potentially incorrect complex structures. In addition, the original authors have not fully resolved issues of data contamination, which further complicates the construction of reliable training sets.

Given these practical and technical limitations, and the fact that executable code for FDA is not publicly available, addressing these issues remains highly challenging. Therefore, in the Discussion section of our revised manuscript, we mention Boltz-2, a framework that integrates structure prediction and binding affinity prediction, as a potential direction for future research.

“I thank the authors for their response and for including two additional baselines, GraphBAN and MONN.

However, the rebuttal does not resolve my primary concern regarding the incomplete baseline comparison. This is essential for substantiating the paper's state-of-the-art (SOTA) claims, and the justifications for excluding key docking-based structure-aware models are not convincing.

1. On the Reproducibility of Baselines

The claim that reproducing Fold-Docking-Affinity (FDA) is "extremely challenging" or "not

practically executable" is inaccurate. The FDA pipeline is a composition of well-known, independently runnable, and publicly available tools: ColabFold (for structure prediction), DiffDock (for docking), and a GNN-based affinity predictor (like GIGN).

Each of these components is functional and widely used. While the authors' specific repository may be for "reference purposes only," the methodology itself is clearly defined and reproducible. Similarly, for Boltz-2, even if training the full structural module is infeasible, it is certainly feasible to retrain their affinity module using the authors' datasets. This would provide a crucial reference point.

2. On the "Task Definition"

The authors' main defense is to state their "task is strictly defined as CPI prediction from protein sequences and compound SMILES only." This definition is precisely the problem. It is an artificial constraint that allows the paper to ignore the most significant recent advances in the field. The current SOTA for this exact problem (predicting affinity from protein sequence and SMILES) involves generating binding structures as an intermediate step.

My original point, which Reviewer #1 also raised, is that a SOTA claim is meaningless if it does not compare against methods that use these computed structures. It is better to know how GenSPARC performs against a baseline that first computes binding poses (e.g., via Boltz-2 or an FDA-like pipeline) and then predicts affinity.

Claiming this is a "different task" is a weak defense. It is the same input-output problem; these docking-based pipelines simply represent the true SOTA competitors. I appreciate that contact-map prediction can be tricky (e.g., with data-leakage risks). Even so, for the affinity-prediction benchmarks, including these baselines seems important. To support the SOTA claims, I'd encourage adding a comparison against models that leverage computed binding (such as Boltz-2 or FDA) for the affinity tasks."

We thank the reviewer for the detailed feedback. While Fold-Docking-Affinity (FDA) is composed of ColabFold, DiffDock, and a GNN-based affinity predictor such as GIGN, reproducing FDA for the datasets used in our study is not straightforward. Accurate reproduction requires not only connecting these components but also re-implementing the specific scoring strategies considered in the original work and retraining for the CPI datasets in our study. Given that executable code for these steps is not publicly available, reproduction is extremely challenging. In addition, the original authors have acknowledged ongoing issues, such as data contamination, which further complicate replication.

Similarly, for Boltz-2, particularly for the affinity prediction component, neither the training code, curated training datasets, nor the data curation procedures have been publicly released. Therefore, it is currently not feasible to retrain or apply Boltz-2's affinity module on our datasets.

Given these limitations, in the Discussion section of our revised manuscript, we mention Boltz-2, a framework that integrates structure prediction and binding affinity prediction, as a potential direction for future research.